# Fox dietary ecology as a tracer of human impact on Pleistocene ecosystems

**Chris Baumann** [1,2,3]*, **Hervé Bocherens**[2,3], **Dorothée G. Drucker**[2], **Nicholas J. Conard**[1,4,5]

**1** Institute for Scientific Archaeology, University of Tübingen, Tübingen, Germany, **2** Senckenberg Centre for Human Evolution and Palaeoenvironment, University of Tübingen, Tübingen, Germany, **3** Biogeology, Department of Geosciences, University of Tübingen, Tübingen, Germany, **4** Department for Early Prehistory and Quaternary Ecology, University of Tübingen, Tübingen, Germany, **5** Senckenberg Centre for Human Evolution and Paleoenvironment, Schloss Hohentübingen, University of Tübingen, Tübingen, Germany

* chris.baumann@uni-tuebingen.de

**Data Availability Statement:** All relevant data are within the paper and its Supporting Information files.

**Funding:** Our research presented received partial financial support by a UNESCO World Cultural

## Abstract

Nowadays, opportunistic small predators, such as foxes (*Vulpes vulpes* and *Vulpes lagopus*), are well known to be very adaptable to human modified ecosystems. However, the timing of the start of this phenomenon in terms of human impact on ecosystems and of the implications for foxes has hardly been studied. We hypothesize that foxes can be used as an indicator of past human impact on ecosystems, as a reflection of population densities and consequently to track back the influence of humans on the Pleistocene environment. To test this hypothesis, we used stable isotope analysis ($\delta^{13}$C, $\delta^{15}$N) of bone collagen extracted from faunal remains from several archaeological sites located in the Swabian Jura (southwest Germany) and covering a time range over three important cultural periods, namely the Middle Palaeolithic (older than 42,000 years ago) attributed to Neanderthals, and the early Upper Palaeolithic periods Aurignacian and Gravettian (42,000 to 30,000 years ago) attributed to modern humans. We then ran Bayesian statistic systems (SIBER, mixSIAR) to reconstruct the trophic niches and diets of Pleistocene foxes. We observed that during the Middle Palaeolithic period, when Neanderthals sparsely populated the Swabian Jura, the niches occupied by foxes suggest a natural trophic behavior. In contrast, during the early Upper Palaeolithic periods, a new trophic fox niche appeared, characterized by a restricted diet on reindeer. This trophic niche could be due to the consumption of human subsidies related to a higher human population density and the resulting higher impact on the Pleistocene environment by modern humans compared to Neanderthals. Furthermore, our study suggests that, a synanthropic commensal behavior of foxes started already in the Aurignacian, around 42,000 years ago.

## Introduction

As soon as hominins started to kill large herbivores, around 2.5 Ma, they started a cascade of ecological reactions that led to vegetation and climate change [1, 2]. Late Pleistocene herbivorous megafauna extinction have been suggested to be at least partially caused by human impact

Heritage doctoral fellowship (CB) funded by Alb-Donau County and the Heritage Authority of Baden-Württemberg (https://www.denkmalpflege-bw.de/), the Senckenberg Nature Research Society (https://www.senckenberg.de/de/) and the University of Tübingen (https://uni-tuebingen.de/). We acknowledge support by Open Access Publishing Fund of University of Tübingen (https://uni-tuebingen.de/). The funders did not play any role in the study design, data collection and analysis, decision to publish, or preparation of the manuscript.

**Competing interests:** The authors have declared that no competing interests exist.

[3, 4]. The impact of human on these large herbivores during the late Quaternary has been largely explored, but less is known about human influence on the ecology of carnivores. Some large carnivorous species may also have been impacted by hominin activities, leading to their extinction, through competition or extirpation [5–7]. However, one aspect that has been rarely addressed is the possibility that human hunting may have had a positive effect on some opportunistic species, through the subsidies that humans produced and that could have been exploited by some species. Especially small predators such as foxes could be one of these cases, considering the ability that both Arctic and red foxes (*Vulpes lagopus* and *Vulpes vulpes*) have to exploit the leftovers of other predators, including humans [8–14]. It is interesting to note that fox remains are often found in archaeological sites of the Late Pleistocene all over Europe [15–22]. To test the hypothesis that fox diet could have been influenced by subsidies from prehistoric hunter-gatherers, we used stable isotopic tracking of bone collagen in Middle and Upper Palaeolithic fossil bones from the Swabian Jura (southwestern Germany), documenting the replacement of Neanderthals (Middle Palaeolithic) by modern humans (Upper Palaeolithic).

The archaeological cave sites of the Swabian Jura, in particular the sites of the Ach and Lone valleys, are among the best scientifically investigated sites of the last glacial in Germany. Especially, the Middle and early Upper Palaeolithic layers (older than 42,000 to 30,000 years ago) contain important lithic and faunal assemblages [20, 23–25]. While during the Middle Paleolithic (from early to middle Würmian period and older than 42,000 years [26–28]) human occupation was spare in this region, it increased during the early Upper Palaeolithic [23]. The early Upper Palaeolithic is represented by the Aurignacian, dated from 42,000 to 34,000 cal BP [28–31] and the Gravettian, dated from 34,000 to 30,000 cal BP [28, 32, 33] in the Swabian Jura. Most of the pre-LGM sites (periods preceding the last glacial maximum) in the Swabian Jura are dominated by faunal remains of cave bear and ungulates [20, 34–41]. The relative abundance of skeletal remains of different species does not necessarily reflect the intensity of faunal exploitation. Niven [42] explained that mainly smaller ungulates, such as reindeer, were taken to the cave by humans as a whole, whereas large animals, such as mammoths, were butchered directly at the kill sites and only certain parts were transported to the cave. In return, this can explain why ivory was often found, but complete mammoth skeletons are missing. Remains of cave bears, which usually have died naturally in the caves during hibernation, do not necessarily relate to human activity, even if there are indications of cave bear hunting [35–37].

In addition to the herbivores, which accounted for the majority of the prey hunted by humans, remains of large and small carnivores have also been found in all of the sites [20, 34–41]. Remains of red foxes and Arctic foxes are particularly common among carnivores and increased from the Middle to the Upper Palaeolithic [17, 20]. The increasing occurrence of fox remains in the early Upper Palaeolithic layers can be explained to some extent by the behavior of foxes. Red and Arctic foxes are generally opportunists, using the food that is most easily accessible to them [43, 44]. Studies of modern red and Arctic foxes have shown that the closer they live to towns or villages, the more they feed on human food leftovers [8, 14, 45, 46]. However, this commensal behavior is not only shown in connection with humans, but also with large carnivores, such as bears and wolves [43, 44, 47, 48]. Without the influence of large predators or humans, both fox species feed mainly on small mammals [8, 9, 11, 12, 43, 44, 46, 48–50]. Studies on Late Pleistocene red and Arctic foxes from Belgium showed that they were slightly larger than today's foxes and lived much more carnivorous [15, 16, 51]. In addition, Szuma and Germonpré [15] concluded that Pleistocene foxes were more adapted to scavenging and thus were more likely commensal to large carnivores or even humans. Consequently,

foxes could also have benefited from the prey of other predators during the Middle and early Upper Palaeolithic of the Swabian Jura, whether they were cave lions, brown bears or humans.

In order to understand the dietary behavior of animals, the use of stable carbon and nitrogen isotopes ($\delta^{13}$C, $\delta^{15}$N) from fossil bone collagen have proven to be extremely informative in recent decades [52–59]. In general, the collagen carbon and nitrogen isotopic values are reflecting the protein part of the diet for omnivores [60], and since meat is much higher in protein than plants, the impact of plant food will be negligible. Thus foxes could be treated in isotopic studies as predators, even if they are known to possibly include plant food in their diet [43, 44]. Especially, the $\delta^{15}$N values in collagen are linked to the trophic level and indicate which prey were consumed in which proportions for carnivorous species [52, 61, 62]. Reconstructions of the trophic isospace, a two-dimensional space, based on the $\delta^{13}$C and $\delta^{15}$N values of consumers (e.g., carnivores) and sources (e.g., large herbivores and rodents), is the basis for determining trophic niches and food reconstructions by using Bayesian statistics [63–66]. Such reconstructions were also performed in archaeological sites of the Swabian Jura during the last years [52, 57, 67–69].

In this study, we firstly reconstruct the trophic niche of foxes over three important cultural time ranges, namely the Middle Palaeolithic, Aurignacian and Gravettian. Based on these results, we consider how a potential commensal to human behavior could be demonstrated and used as an indicator of human population densities and consequently to track back the influence of humans on the Pleistocene environment. To find an answer, we reconstructed the trophic niches and diet of Middle Palaeolithic, Aurignacian and Gravettian foxes from the Swabian Jura, based on their $\delta^{13}$C and $\delta^{15}$N isotopic values.

## Material and methods

### Material

In this study, we present 70 new $\delta^{13}$C and $\delta^{15}$N isotopic values of foxes and large carnivores from Middle Palaeolithic, Aurignacian and Gravettian layers of the Ach and Lone valleys (Table 1) as well as a set of 44 new small mammal isotopic values from the same periods (Table 2). No permits were required for the described study, which complied with all relevant regulations. All samples used in this study are stored in the storage facilities of the Institute for Scientific Archaeology (University of Tübingen), headed by Nicholas J. Conard. Preserved collagen samples are stored in the storage facilities of Biogeology (headed and managed by Hervé Bocherens), Department of Geosciences (University of Tübingen).

All newly analyzed specimens were most likely adult, distinguished by symphysial fusing and tooth characteristics. To exclude using samples from the same individual, most carnivore samples came from separate archaeological sites or layers. In total, our samples reflect a minimum of 62 single carnivore specimens (more information in the chapter "intra-individual variability" of S1 Text).

To generate a representative isospace we have added published isotope values of nine large predators and 51 large herbivores [54, 57, 69]. More detailed information is given in S1 Table. In total, we considered seven carnivore species. For the Middle Paleolithic we included wolf (*Canis lupus*), brown bear (*Ursus arctos*), red fox (*Vulpes vulpes*) and Arctic fox (*Vulpes lagopus*). The Aurignacian is represented by wolf, wolverine (*Gulo gulo*), lynx (*Lynx lynx*), hyena (*Crocuta crocuta*), brown bear, red fox and Arctic fox. For the Gravettian, we considered the following carnivores: wolf, wolverine, lynx, cave lion (*Panthera leo spelaea*), brown bear, red fox and Arctic fox. As the isospaces for the different pre-LGM periods are quite similar [57], we can use the complete set of vole (*Microtus arvalis/agrestis*), Norway and Arctic lemmings (*Lemmus lemmus* and *Dicrostonyx torquatus*), horse (*Equus* sp.), hare (*Lepus* sp.), mammoth

**Table 1. List of newly analyzed isotopic values of carnivores reported in this study.**

| Lab ID | Period | Location | Excav. No | AH | Taxon | Skeleton element | $N_{bone}$ [%] | Yield [mg/g] | $C_{coll}$ [%] | $N_{coll}$ [%] | $C/N_{coll}$ | $\delta^{13}C_{coll}$ [‰] | $\delta^{15}N_{coll}$ [‰] |
|---|---|---|---|---|---|---|---|---|---|---|---|---|---|
| PLC-79 | MP | BS | BS 550825–35 | | *Canis lupus* | Radius | 1.7 | 56.0 | 34.0 | 11.8 | 3.4 | -20.1 | 10.6 |
| PLC-35 | MP | HS | HS 18/13-2806 | MP R | *Canis lupus* | Ulna | 2.2 | 64.7 | 35.2 | 12.4 | 3.3 | -20.0 | 7.7 |
| PLC-37 | MP | HS | HS 13/10-8071 | MP U | *Canis lupus* | Mandible | 1.1 | 33.8 | 34.6 | 12.1 | 3.3 | -20.1 | 8.5 |
| PLC-38 | MP | HS | HS 17/11-4075 | MP U | *Canis lupus* | Mandible | 2.1 | 74.4 | 33.6 | 11.8 | 3.3 | -19.4 | 10.0 |
| PLC-48 | MP | VH | Vg VIII 7773 | VIII | *Canis lupus* | Astragalus | 2.4 | 104.5 | 38.8 | 13.6 | 3.3 | -19.2 | 8.3 |
| PLC-49 | MP | VH | Vg VIII 12692 | VIII | *Canis lupus* | Tibia | 2.3 | 119.2 | 42.2 | 14.4 | 3.4 | -19.7 | 7.3 |
| PLC-76 | MP | BS | BS 13.8.34/6 | | *Vulpes lagopus* | Mandible | 0.9 | 27.3 | 40.0 | 13.3 | 3.5 | -21.4 | 1.0 |
| PLC-80 | MP | BS | BS 12.9.34/10 | | *Vulpes vulpes* | Tibia | 0.8 | 41.7 | 26.4 | 9.1 | 3.4 | -19.7 | 10.0 |
| PLC-82 | MP | BS | BS 18.8.34/13 | | *Vulpes vulpes* | Radius | 1.5 | 71.5 | 42.5 | 15.0 | 3.3 | -20.2 | 7.8 |
| PLC-83 | MP | BS | BS 28.8.33/59 | | *Vulpes vulpes* | Humerus | 2.7 | 135.2 | 43.1 | 15.3 | 3.3 | -20.4 | 8.2 |
| PLC-84 | MP | BS | BS 11.9.34/37 | | *Vulpes vulpes* | Mandible | 0.9 | 34.0 | 26.7 | 9.2 | 3.4 | -20.1 | 9.0 |
| VLP-10 | MP | HF | HF 68/2989 | VI | *Vulpes vulpes* | Tibia | 3.7 | 73.2 | 42.0 | 14.9 | 3.3 | -21.1 | 3.0 |
| PLC-39 | MP | HS | HS 13/9-8156 | MP U | *Vulpes vulpes* | Mandible | 2.8 | 142.4 | 39.8 | 14.0 | 3.3 | -20.3 | 8.4 |
| PLC-40 | MP | HS | HS 14/8-10670 | MP U | *Vulpes vulpes* | Ulna | 2.5 | 103.5 | 41.7 | 14.6 | 3.3 | -19.9 | 8.6 |
| PLC-78 | A | BS | BS 24.8.55/22 | | *Canis lupus* | Tibia | 2.9 | 107.0 | 41.0 | 14.6 | 3.3 | -19.5 | 8.5 |
| JK2175 | A | HF | HF 24/1035 | IIIa | *Canis lupus* | Ulna | 2.5 | 67.2 | 34.4 | 12.2 | 3.3 | -19.5 | 10.7 |
| JK2180 | A | HF | HF 89/1553 | IV | *Canis lupus* | Humerus | 3.1 | 110.3 | 40.9 | 14.6 | 3.3 | -18.6 | 8.3 |
| JK2184 | A | HF | HF 79/2563 | IV | *Canis lupus* | Metacarpal IV | 3.4 | 96.8 | 40.2 | 14.3 | 3.3 | -18.9 | 10.0 |
| PLC-24 | A | HS | HS 19/2-9285 | | *Canis lupus* | Mandible | 1.0 | 43.6 | 17.2 | 6.1 | 3.3 | -19.1 | 9.5 |
| PLC-25 | A | HS | HS 19/2-9312 | | *Canis lupus* | Mandible | 0.6 | 14.9 | 34.8 | 12.2 | 3.3 | -19.0 | 9.6 |
| PLC-29 | A | HS | HS 19/6-1435 | | *Canis lupus* | Ulna | 2.8 | 108.4 | 41.8 | 14.8 | 3.3 | -19.4 | 10.7 |
| PLC-30 | A | HS | HS 12/5-8905 | | *Canis lupus* | Atlas | 2.5 | 85.0 | 40.1 | 14.1 | 3.3 | -19.7 | 9.2 |
| PLC-31 | A | HS | HS 19/3-2467 | | *Canis lupus* | Ulna | 1.9 | 74.7 | 42.2 | 14.6 | 3.4 | -19.0 | 9.8 |
| PLC-32 | A | HS | HS 18/4-3805 | | *Canis lupus* | Humerus | 3.2 | 158.8 | 41.9 | 14.8 | 3.3 | -19.5 | 8.5 |
| PLC-2 | A | VH | Vg IV 9059 | IV | *Canis lupus* | Radius | 3.2 | 145.1 | 41.7 | 14.7 | 3.3 | -20.2 | 9.1 |
| PLC-3 | A | VH | Vg IV/V 8200 | IV/V | *Canis lupus* | Atlas | 0.9 | 27.9 | 35.9 | 12.5 | 3.3 | -21.0 | 9.4 |
| PLC-44 | A | VH | Vg V 12645 | V | *Canis lupus* | Metacarpal | 2.8 | 127.0 | 42.8 | 14.9 | 3.3 | -18.9 | 9.6 |
| PLC-45 | A | VH | Vg IV 10685 | IV | *Canis lupus* | Tibia | 1.6 | 63.4 | 41.7 | 14.7 | 3.3 | -20.4 | 9.5 |
| PLC-46 | A | VH | Vg IV 1732 | IV | *Canis lupus* | Ulna | 3.2 | 132.7 | 40.7 | 14.5 | 3.3 | -19.7 | 9.3 |
| PLC-47 | A | VH | Vg IV 7214 | IV | *Canis lupus* | Ulna | 2.3 | 77.4 | 38.1 | 13.4 | 3.3 | -19.2 | 8.9 |

*(Continued)*

**Table 1.** (Continued)

| Lab ID | Period | Location | Excav. No | AH | Taxon | Skeleton element | $N_{bone}$ [%] | Yield [mg/g] | $C_{coll}$ [%] | $N_{coll}$ [%] | $C/N_{coll}$ | $\delta^{13}C_{coll}$ [‰] | $\delta^{15}N_{coll}$ [‰] |
|---|---|---|---|---|---|---|---|---|---|---|---|---|---|
| PLC-62 | A | Si | Si 1631 | IV | *Gulo gulo* | Femur | 2.1 | 67.3 | 34.8 | 12.4 | 3.3 | -19.1 | 9.4 |
| PLC-17 | A | GK | GK 69/540 | IIb | *Lynx lynx* | Phalanx | | 117.6 | 40.0 | 14.1 | 3.3 | -19.3 | 7.7 |
| PLC-23 | A | HS | HS 18/7-11629 | | *Lynx lynx* | Humerus | 2.6 | 79.8 | 40.4 | 14.2 | 3.3 | -19.3 | 10.2 |
| PLC-63 | A | Si | Si 3892 | | *Lynx lynx* | Tibia | 2.9 | 132.7 | 43.2 | 15.1 | 3.3 | -19.8 | 7.0 |
| VLP-1 | A | GK | GK 35/206 | III | *Vulpes lagopus* | Tibia | 3.1 | 105.3 | 43.8 | 15.4 | 3.3 | -20.6 | 9.1 |
| VLP-3 | A | HF | HF 25/1111 | VAA | *Vulpes lagopus* | Radius | 2.8 | 76.0 | 43.1 | 15.1 | 3.3 | -19.9 | 8.6 |
| PLC-22 | A | HS | HS 17/4-5119 | | *Vulpes lagopus* | Mandible | 2.9 | 133.8 | 42.8 | 14.7 | 3.4 | -20.2 | 5.4 |
| PLC-28 | A | HS | HS 19/7-11526 | | *Vulpes lagopus* | Mandible | 2.1 | 62.9 | 35.9 | 12.6 | 3.3 | -19.7 | 8.6 |
| PLC-55 | A | HS | HS 17/7 7067 | | *Vulpes lagopus* | Mandible | 1.6 | 31.2 | 16.9 | 6.1 | 3.2 | -20.1 | 8.9 |
| PLC-1 | A | VH | Vg IV 7213 | IV | *Vulpes lagopus* | Tibia | 2.6 | 113.6 | 41.9 | 14.6 | 3.4 | -18.4 | 8.7 |
| PLC-16 | A | VH | Vg IV 245 | IV | *Vulpes lagopus* | Mandible | 1.2 | 50.6 | 32.7 | 11.6 | 3.3 | -19.4 | 9.4 |
| PLC-15 | A | VH | Vg IV 12782 | IV | *Vulpes* sp. | Mandible | 1.4 | 56.7 | 35.1 | 12.2 | 3.4 | -20.4 | 8.9 |
| PLC-85 | A | BS | BS 34/19 | | *Vulpes vulpes* | Tibia | 2.5 | 112.4 | 43.9 | 15.5 | 3.3 | -20.2 | 8.3 |
| PLC-26 | A | HS | HS 19/2-9298 | | *Vulpes vulpes* | Mandible | 0.8 | 30.9 | 27.2 | 9.6 | 3.3 | -19.8 | 8.1 |
| PLC-27 | A | HS | HS 19/2-9359 | | *Vulpes vulpes* | Humerus | 0.6 | 23.9 | 17.2 | 6.2 | 3.2 | -20.2 | 8.2 |
| PLC-66 | A | Si | Si 3360 | IV | *Vulpes vulpes* | Mandible | 1.8 | 76.3 | 40.5 | 13.8 | 3.4 | -20.3 | 8.0 |
| PLC-67 | A | Si | Si 3361 | IV | *Vulpes vulpes* | Mandible | 2.6 | 143.8 | 42.2 | 14.8 | 3.3 | -21.0 | 8.2 |
| PLC-68 | A | Si | Si 3448 | IV | *Vulpes vulpes* | Humerus | 2.7 | 138.6 | 42.6 | 14.8 | 3.4 | -20.4 | 6.0 |
| PLC-69 | A | Si | Si 3446 | IV | *Vulpes vulpes* | Tibia | 3.2 | 150.0 | 43.4 | 15.1 | 3.4 | -20.0 | 4.8 |
| PLC-10 | A | VH | Vg IV 7245 | IV | *Vulpes vulpes* | Tibia | 1.1 | 36.2 | 37.7 | 13.1 | 3.4 | -19.2 | 8.2 |
| PLC-11 | A | VH | Vg IV 7259 | IV | *Vulpes vulpes* | Radius | 3.2 | 148.2 | 42.3 | 14.9 | 3.3 | -19.6 | 9.1 |
| PLC-13 | A | VH | Vg IV 12776 | IV | *Vulpes vulpes* | Mandible | 2.3 | 112.6 | 38.9 | 13.6 | 3.3 | -20.0 | 4.7 |
| PLC-14 | A | VH | Vg IV 12780 | IV | *Vulpes vulpes* | Mandible | 1.1 | 42.8 | 34.1 | 11.9 | 3.4 | -20.0 | 5.5 |
| PLC-8 | A | VH | Vg IV/V 11675 | IV/V | *Vulpes vulpes* | Femur | 2.8 | 58.4 | 40.7 | 14.4 | 3.3 | -20.1 | 8.3 |
| PLC-9 | A | VH | Vg IV 3551 | IV | *Vulpes vulpes* | Tibia | 1.6 | 44.7 | 31.6 | 11.2 | 3.3 | -19.4 | 5.7 |
| JK2174 | G | HF | HF 56/1965 | IIC | *Canis lupus* | Scapula | 3.5 | 144.8 | 39.6 | 14.1 | 3.3 | -20.2 | 9.7 |
| JK2183 | G | HF | HF 59/1390 | IIcf | *Canis lupus* | Calcaneus | 3.4 | 155.7 | 40.9 | 14.4 | 3.3 | -20.2 | 9.3 |
| JK2178 | G | HF | HF 99/1174 | IIC | *Canis lupus* | Metacarpale II | 3.7 | 11.9 | 40.6 | 14.5 | 3.3 | -19.5 | 8.9 |
| PLC-70 | G | Si | Si 983 | I | *Gulo gulo* | Scapula | 3.1 | 165.1 | 41.7 | 14.8 | 3.3 | -19.1 | 7.6 |
| PLC-18 | G | GK | GK 9/3 | Ir | *Lynx lynx* | Rib | 2.0 | 96.2 | 42.5 | 14.8 | 3.3 | -19.5 | 8.4 |

(*Continued*)

**Table 1.** (Continued)

| Lab ID | Period | Location | Excav. No | AH | Taxon | Skeleton element | $N_{bone}$ [%] | Yield [mg/g] | $C_{coll}$ [%] | $N_{coll}$ [%] | $C/N_{coll}$ | $\delta^{13}C_{coll}$ [‰] | $\delta^{15}N_{coll}$ [‰] |
|---|---|---|---|---|---|---|---|---|---|---|---|---|---|
| PLC-19 | G | GK | GK 121/93 | Ir | *Lynx lynx* | Mandible | 3.6 | 137.2 | 42.6 | 14.8 | 3.3 | -18.7 | 8.0 |
| PLC-77 | G | BS | BS 24.9.53/15 | | *Vulpes lagopus* | Ulna | 2.2 | 97.6 | 41.6 | 14.7 | 3.3 | -20.1 | 8.7 |
| PLC-42 | G | Si | Si 776 | I | *Vulpes lagopus* | Mandible | 2.8 | 138.2 | 42.4 | 14.8 | 3.3 | -20.3 | 7.6 |
| VLP-4 | G | GK | GK 508/70 | I | *Vulpes vulpes* | Tibia | 3.2 | 109.8 | 44.2 | 15.3 | 3.4 | -19.7 | 7.1 |
| VLP-5 | G | GK | GK 15/106 | I | *Vulpes vulpes* | Tibia | 3.1 | 109.7 | 44.3 | 15.3 | 3.4 | -19.7 | 9.7 |
| PLC-43 | G | Si | Si 773 | I | *Vulpes vulpes* | Mandible | 3.2 | 166.9 | 43.9 | 15.1 | 3.4 | -19.7 | 4.0 |
| PLC-71 | G | Si | Si 2862 | I | *Vulpes vulpes* | Humerus | 3.1 | 157.9 | 43.0 | 14.7 | 3.4 | -20.5 | 6.0 |
| PLC-72 | G | Si | Si 2550 | I | *Vulpes vulpes* | Mandible | 2.6 | 139.4 | 42.5 | 14.7 | 3.4 | -19.4 | 6.7 |
| PLC-73 | G | Si | Si 2214 | I | *Vulpes vulpes* | Humerus | 2.8 | 135.5 | 44.0 | 14.9 | 3.4 | -19.6 | 3.7 |
| PLC-75 | G | Si | Si 2213 | I | *Vulpes vulpes* | Tibia | 1.5 | 54.4 | 32.3 | 11.2 | 3.4 | -20.3 | 9.2 |

AH = Archaeological horizon, MP = Middle Palaeolithic, A = Aurignacian, G = Gravettian, BS = Bockstein, HS = Hohlenstein-Stadel, HF = Hohle Fels, GK = Geißenklösterle, Si = Sirgenstein, VH = Vogelherd.

(*Mammuthus primigenius*) and reindeer (*Rangifer tarandus*) samples as dietary sources to reconstruct the trophic niches of the carnivores. All of our studied material come from archaeological cave sites from the Ach Valley (Hohle Fels, Geißenklösterle and Sirgenstein) and the Lone Valley (Bockstein, Hohlenstein-Stadel and Vogelherd) (Fig 1).

The taxonomic determination of carnivore specimens was done following published morphological and metrical studies [24, 38, 40, 41, 70], as well as by comparing the bones with the zooarchaeological collection of the University of Tübingen. However, in the rest of study, we will combine red fox and Arctic fox as "fox", since they do not show a clear trophic niche differentiation in the Middle and Upper Palaeolithic of the Swabian Jura [67, 71] (see chapter "Statistical test for isotopic variance of both fox species" in S2 Text). The newly analyzed small mammals were determined by using published determination keys [72–74].

### Elemental and isotope analyses

For the isotopic analysis of the larger bones (Lab codes: JK, PLC, VLP), bone samples (0.3–0.7 g) were cut using a Saeshin Forte 200 alpha micro-circular saw. After successive cleaning in Millipore water and acetone, the samples were ground to powder manually (grain size less than 0.7 mm). In the case of the rodent samples (Lab code: SJM), the complete mandible without teeth was taken for each specimen and grinded manually with a mortar, which resulted in a smaller grain size of the samples, but in a higher yield of bone powder. The collagen content of the bone was only measured for JK, PLC and VLP samples by performing a CNS elemental analysis following Bocherens [55]. This analysis was performed at the Hydrogeochemisty working group (University of Tübingen) using a Vario EL elemental analyzer. Sulfanilic acid from Merck was used as the international standard. The SJM samples did not have enough material to perform this preliminary analysis and were run directly for collagen extraction.

**Table 2.  List of newly analyzed isotopic values of rodents and hare reported in this study.**

| Lab ID | Period | Location | Excav. No | AH | Taxon | Element | $N_{bone}$ [%] | yield [mg/g] | $C_{coll}$ [%] | $N_{coll}$ [%] | C/ $N_{coll}$ | $\delta^{13}C_{coll}$ [‰] | $\delta^{15}N_{coll}$ [‰] |
|---|---|---|---|---|---|---|---|---|---|---|---|---|---|
| SJM-54 | A | HF | HF Eimer-1004 (Qu 11, AH Vab) | AH Vab | Dicrostonyx sp. | Mandible | | 33.4 | 31.3 | 11.3 | 3.2 | -21.2 | 6.4 |
| SJM-55 | A | HF | HF Eimer-1004 (Qu 11, AH Vab) | AH Vab | Dicrostonyx sp. | Mandible | | 41.5 | 29.5 | 10.6 | 3.2 | -20.3 | 7.1 |
| SJM-56 | A | HF | HF Eimer-1002 (Qu 11, AH Vab) | AH Vab | Dicrostonyx sp. | Mandible | | 68.6 | 23.9 | 8.7 | 3.2 | -20.9 | 5.2 |
| SJM-57 | A | HF | HF Eimer-1004 (Qu 11, AH Vab) | AH Vab | Lemmus lemmus | Mandible | | 51.0 | 36.8 | 13.0 | 3.3 | -21.4 | 5.2 |
| SJM-58 | A | HF | HF Eimer-1004 (Qu 11, AH Vab) | AH Vab | Lemmus lemmus | Mandible | | 47.6 | 27.7 | 9.7 | 3.3 | -22.8 | 6.6 |
| SJM-59 | A | HF | HF Eimer-1002 (Qu 11, AH Vab) | AH Vab | Lemmus lemmus | Mandible | | 59.5 | 28.4 | 10.1 | 3.3 | -21.6 | 3.9 |
| SJM-60 | A | HF | HF Eimer-1002 (Qu 11, AH Vab) | AH Vab | Lemmus lemmus | Mandible | | 57.5 | 30.1 | 10.6 | 3.3 | -20.9 | 5.6 |
| SJM-61 | A | HF | HF Eimer-1002 (Qu 11, AH Vab) | AH Vab | Lemmus lemmus | Mandible | | 51.9 | 32.2 | 11.5 | 3.3 | -21.7 | 4.2 |
| SJM-62 | A | HF | HF Eimer-719 (Qu 32, AH IV) | AH IV | Microtus arvalis/ agrestis | Mandible | | 31.2 | 21.9 | 7.6 | 3.4 | -22.1 | 3.8 |
| SJM-63 | A | HF | HF Eimer-1277 (Qu 31, AH IV) | AH IV | Microtus arvalis/ agrestis | Mandible | | 41.3 | 23.3 | 8.0 | 3.4 | -21.8 | 4.5 |
| SJM-50 | G | HF | HF Eimer-1225 (Qu 110, AH IIc) | AH IIc | Dicrostonyx sp. | Mandible | | 64.5 | 31.9 | 11.5 | 3.2 | -21.1 | 3.6 |
| SJM-51 | G | HF | HF Eimer-1225 (Qu 110, AH IIc) | AH IIc | Dicrostonyx sp. | Mandible | | 75.6 | 32.4 | 11.4 | 3.3 | -21.3 | 6.1 |
| SJM-52 | G | HF | HF Eimer-1429 (Qu 112, AH IIc) | AH IIc | Dicrostonyx sp. | Mandible | | 71.8 | 34.7 | 12.3 | 3.3 | -20.2 | 3.5 |
| VLP-12 | G | GK | GK 99/458 | It | Lepus sp. | Tibia | 3.0 | 92.2 | 43.4 | 15.3 | 3.3 | -20.4 | 2.8 |
| VLP-13 | G | GK | GK 86/17 | Ir | Lepus sp. | Tibia | 2.6 | 59.1 | 43.6 | 15.5 | 3.3 | -20.2 | 3.5 |
| SJM-53 | G | HF | HF Eimer-1225 (Qu 110, AH IIc) | AH IIc | Microtus arvalis/ agrestis | Mandible | | 20.4 | 30.9 | 11.1 | 3.2 | -21.7 | 3.0 |
| SJM-7 | MP | HF | HF Eimer-1613 (Qu 26, AH IX WF) | AH IX WF | Dicrostonyx sp. | Mandible | | 61.0 | 33.1 | 11.7 | 3.3 | -21.2 | 5.5 |
| SJM-8 | MP | HF | HF Eimer-1613 (Qu 26, AH IX WF) | AH IX WF | Dicrostonyx sp. | Mandible | | 42.1 | 29.7 | 10.7 | 3.2 | -21.0 | 6.7 |
| SJM-9 | MP | HF | HF Eimer-1613 (Qu 26, AH IX WF) | AH IX WF | Dicrostonyx sp. | Mandible | | 70.1 | 32.9 | 11.6 | 3.3 | -21.2 | 7.1 |
| SJM-11 | MP | HF | HF Eimer-1613 (Qu 26, AH IX WF) | AH IX WF | Dicrostonyx sp. | Mandible | | 86.8 | 25.1 | 8.0 | 3.6 | -25.1 | -1.5 |
| SJM-1 | MP | HF | HF Eimer-1613 (Qu 26, AH IX WF) | AH IX WF | Lemmus lemmus | Mandible | | 52.3 | 34.7 | 12.2 | 3.3 | -21.8 | 5.9 |
| SJM-2 | MP | HF | HF Eimer-1613 (Qu 26, AH IX WF) | AH IX WF | Lemmus lemmus | Mandible | | 60.1 | 33.6 | 11.8 | 3.3 | -22.2 | 4.8 |
| SJM-3 | MP | HF | HF Eimer-1613 (Qu 26, AH IX WF) | AH IX WF | Lemmus lemmus | Mandible | | 44.8 | 32.7 | 11.5 | 3.3 | -20.7 | 6.5 |
| SJM-4 | MP | HF | HF Eimer-1613 (Qu 26, AH IX WF) | AH IX WF | Lemmus lemmus | Mandible | | 55.4 | 35.4 | 12.2 | 3.4 | -22.0 | 5.8 |
| SJM-5 | MP | HF | HF Eimer-1613 (Qu 26, AH IX WF) | AH IX WF | Lemmus lemmus | Mandible | | 69.5 | 33.8 | 11.7 | 3.4 | -21.9 | 7.3 |
| SJM-6 | MP | HF | HF Eimer-1613 (Qu 26, AH IX WF) | AH IX WF | Lemmus lemmus | Mandible | | 72.0 | 32.0 | 11.4 | 3.3 | -21.8 | 6.8 |
| SJM-12 | MP | HF | HF Eimer-1613 (Qu 26, AH IX WF) | AH IX WF | Lemmus lemmus | Mandible | | 54.7 | 30.9 | 11.3 | 3.2 | -23.1 | 1.0 |

*(Continued)*

**Table 2.** (Continued)

| Lab ID | Period | Location | Excav. No | AH | Taxon | Element | $N_{bone}$ [%] | yield [mg/g] | $C_{coll}$ [%] | $N_{coll}$ [%] | C/ $N_{coll}$ | $\delta^{13}C_{coll}$ [‰] | $\delta^{15}N_{coll}$ [‰] |
|---|---|---|---|---|---|---|---|---|---|---|---|---|---|
| SJM-13 | MP | HF | HF Eimer-1613 (Qu 26, AH IX WF) | AH IX WF | Lemmus lemmus | Mandible | | 70.3 | 31.1 | 11.3 | 3.2 | -23.2 | 5.2 |
| SJM-14 | MP | HF | HF Eimer-1613 (Qu 26, AH IX WF) | AH IX WF | Lemmus lemmus | Mandible | | 68.0 | 25.3 | 9.2 | 3.2 | -24.9 | -0.6 |
| SJM-15 | MP | HF | HF Eimer-1613 (Qu 26, AH IX WF) | AH IX WF | Lemmus lemmus | Mandible | | 86.0 | 27.7 | 9.3 | 3.5 | -24.6 | 1.0 |
| SJM-16 | MP | HF | HF Eimer-1613 (Qu 26, AH IX WF) | AH IX WF | Lemmus lemmus | Mandible | | 70.5 | 26.0 | 8.4 | 3.6 | -24.5 | 0.6 |
| SJM-17 | MP | HF | HF Eimer-1613 (Qu 26, AH IX WF) | AH IX WF | Lemmus lemmus | Mandible | | 79.3 | 28.5 | 10.5 | 3.2 | -22.9 | 2.5 |
| SJM-18 | MP | HF | HF Eimer-1613 (Qu 26, AH IX WF) | AH IX WF | Lemmus lemmus | Mandible | | 60.2 | 26.7 | 8.9 | 3.5 | -23.5 | 2.5 |
| SJM-10 | MP | HF | HF Eimer-1613 (Qu 26, AH IX WF) | AH IX WF | Microtus arvalis/ agrestis | Mandible | | 63.1 | 30.4 | 10.9 | 3.3 | -22.4 | 7.3 |
| SJM-29 | MP | HF | HF Eimer-1613 (Qu 26, AH IX WF) | AH IX WF | Microtus arvalis/ agrestis | Mandible | | 44.3 | 27.8 | 9.4 | 3.5 | -22.5 | 7.9 |
| SJM-30 | MP | HF | HF Eimer-1613 (Qu 26, AH IX WF) | AH IX WF | Microtus arvalis/ agrestis | Mandible | | 48.2 | 25.7 | 9.0 | 3.3 | -22.1 | 4.5 |
| SJM-31 | MP | HF | HF Eimer-1613 (Qu 26, AH IX WF) | AH IX WF | Microtus arvalis/ agrestis | Mandible | | 30.1 | 25.2 | 8.6 | 3.4 | -22.6 | 6.7 |
| SJM-32 | MP | HF | HF Eimer-1613 (Qu 26, AH IX WF) | AH IX WF | Microtus arvalis/ agrestis | Mandible | | 59.3 | 26.9 | 9.2 | 3.4 | -23.0 | 7.4 |
| SJM-33 | MP | HF | HF Eimer-1613 (Qu 26, AH IX WF) | AH IX WF | Microtus arvalis/ agrestis | Mandible | | 45.1 | 25.7 | 8.7 | 3.4 | -23.1 | 5.4 |
| SJM-34 | MP | HF | HF Eimer-1613 (Qu 26, AH IX WF) | AH IX WF | Microtus arvalis/ agrestis | Mandible | | 67.1 | 21.4 | 7.9 | 3.2 | -22.0 | 5.7 |
| SJM-35 | MP | HF | HF Eimer-1613 (Qu 26, AH IX WF) | AH IX WF | Microtus arvalis/ agrestis | Mandible | | 47.3 | 25.1 | 8.8 | 3.3 | -22.7 | 4.4 |
| SJM-36 | MP | HF | HF Eimer-1613 (Qu 26, AH IX WF) | AH IX WF | Microtus arvalis/ agrestis | Mandible | | 59.3 | 23.2 | 7.9 | 3.5 | -22.2 | 5.9 |
| SJM-37 | MP | HF | HF Eimer-1613 (Qu 26, AH IX WF) | AH IX WF | Microtus arvalis/ agrestis | Mandible | | 65.6 | 22.6 | 8.2 | 3.2 | -21.8 | 5.7 |
| SJM-38 | MP | HF | HF Eimer-1613 (Qu 26, AH IX WF) | AH IX WF | Microtus arvalis/ agrestis | Mandible | | 63.1 | 23.4 | 8.4 | 3.2 | -23.0 | 6.2 |

AH = Archaeological horizon.

Collagen extraction following the protocol of Bocherens [53] was performed in the Biogeology working group (University of Tübingen). Depending on the percentage of nitrogen in the bone powder (%$N_{bone}$) of each sample, as measured by the CNS analysis, we used 120 mg (4.0–4.5%$N_{bone}$) to 450 mg (0.4–1.0%$N_{bone}$) of bone powder for the extraction. In the case of the SJM samples we used the totality of the available powder, because the average sample size was only 40 mg. With respect to the smaller grain size of the SJM samples we have reduced the time in which the sample remains in 1 molar HCl solution from the 20 minutes recommended in the protocol to 15 minutes to avoid collagen damage for fine grain bone powder. The collagen extraction process included a step of soaking the bone powder in 0.125 M NaOH between the demineralization and gelatinization steps to achieve the elimination of lipids and humic acids. After this process, the samples were freeze-dried.

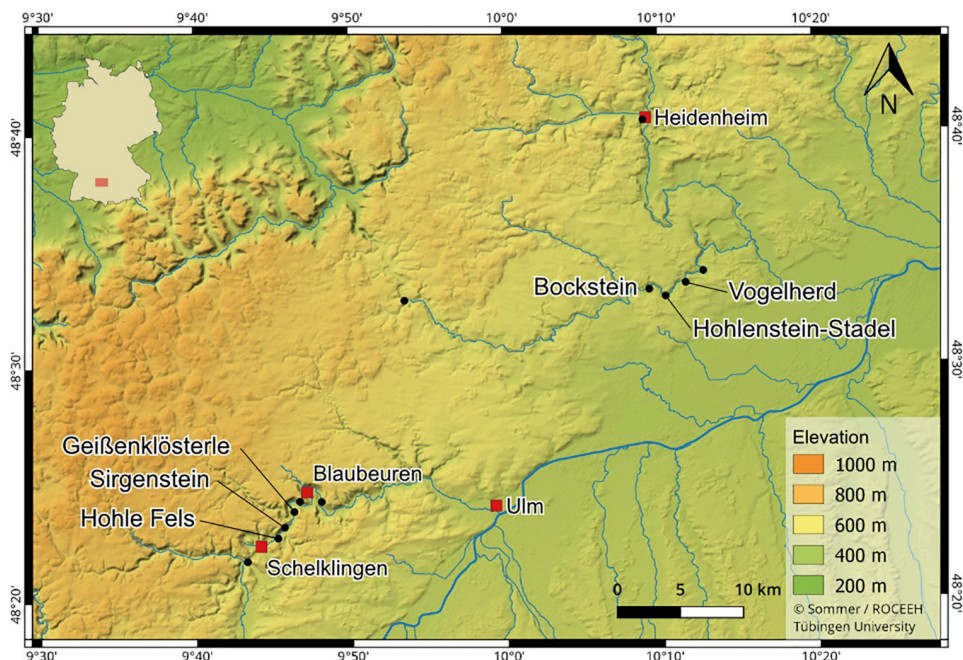

**Fig 1. Map of the studied sites.** Map of sites included in this study. https://doi.org/10.5281/zenodo.3460300 (CC BY 4.0 license).

The elemental analyses ($C_{coll}$, $N_{coll}$) and isotopic measurements of collagen ($\delta^{13}C_{coll}$, $\delta^{15}N_{coll}$) for the VLP samples (n = 7) were performed at the Geochemical department of the University of Tübingen, using an elemental analyzer NC 2500 connected to a Thermo Quest Delta+XL mass spectrometer. While the elemental analyses and isotopic measurements for JK (n = 8), 38 PLC samples (PLC-1 –PLC-49) and 10 SJM samples (SJM-1 –SJM-10) were undertaken at the Laboratory of Chronology (Finnish Museum of Natural History), using an NC 2500 elemental analyzer coupled to a Thermo Delta V Plus isotope ratio mass spectrometer. For the 21 PLC samples (PLC-55 –PLC-85), and the 32 SJM samples (SJM-11 –SJM-63), the elemental analyses and isotopic measurements were performed in duplicate at the Institute of Environmental Science and Technology of the Universitat Autònoma de Barcelona (ICTA-UAB) using a Thermo Flash 1112 (Thermo ScientificVC) elemental analyzer coupled to a Thermo Delta V Advantage mass spectrometer with a Conflo III interface.

The samples analyzed in Tübingen and Helsinki were calibrated to $\delta^{13}C$ values of USGS-40 ($\delta^{13}C$ = -26.8‰, $\delta^{15}N$ = -4.7‰) and USGS-41 ($\delta^{13}C$ = +36.1‰, $\delta^{15}N$ = +46.7‰). Based on multiple measurements of matrix matched in-house reference materials, the reproducibility was ±0.19‰ for $\delta^{13}C$, ±0.24‰ for $\delta^{15}N$ values. The reproducibility error for the amounts of C and N was lower than 4%.

At the laboratory of Barcelona, the international laboratory standard IAEA 600 (caffeine) was used as well as two in-house reference materials (modern collagen of camel and elk). These same two in-house reference materials were also used for the isotopic analyses performed in Tübingen and Helsinki, assuring the comparability of all measured isotopic values independently of where they were performed. An analytical error below 0.2‰ (1σ) was determined for $\delta^{13}C$ and $\delta^{15}N$ in all the repeated analyses. The reproducibility error for the amounts of C and N was lower than 2%.

Following the recommendations of DeNiro [75] and Ambrose [76], we only used collagen samples with a carbon-to-nitrogen-ratio ($C:N_{coll}$) between 2.9 and 3.6 and a percentage of nitrogen higher than 5% for palaeoecological interpretations.

All commissioned laboratories measure the ratios of $^{13}C/^{12}C$ and $^{15}N/^{14}N$ relative to a standard (V-PDB for carbon and AIR for nitrogen). The isotopic ratios are expressed using the $\delta$ (delta) value as follows:

$$\delta^{13}C = \left[ (^{13}C/^{12}C)_{sample}/(^{13}C/^{12}C)_{reference} - 1 \right] \text{ x } 1000 \text{ (‰)}$$

$$\delta^{15}N = \left[ (^{15}N/^{14}N)_{sample}/(^{15}N/^{14}N)_{reference} - 1 \right] \text{ x } 1000 \text{ (‰)}$$

## Trophic niche modeling

To reconstruct the fox trophic niches, we first applied a multivariate cluster analysis to the $\delta^{13}C$ and $\delta^{15}N$ isotopic values in JMP 14 with respect to the stratigraphic association, namely Middle Palaeolithic, Aurignacian and Gravettian. As a result, we obtained different clusters for each of the periods. We then used the R package SIBER (Stable Isotope Bayesian Ellipses in R) to calibrate the niches out of the clusters [65]. It was possible to reconstruct the complete niches (= convex hull or total area, TA, Layman [77]) that include all members of the clusters, given by the isotopic values of our samples. Furthermore, we calculated the core niches (= standard ellipse area, SEA, Jackson [65]) that explain 40% of all potential specimens that will fit into these niches, based on a most likelihood estimation in a Bayesian framework. While the complete niche is quite sensitive to the given sample size, the core niche is more reliable for analyzing small assemblages and is recommend by Jackson [65]. To examine the trophic niche overlap between foxes and large carnivores, we calculated, additionally to SEA and TA, the standard ellipse area corrected for sample size (SEAc). Based on this, the percentage of overlap in the respective core niches could be estimated. Throughout this study, we use the term "niche" for a trophic niche.

## Dietary reconstruction

To build the isospace for the dietary reconstruction, we used prey groups, combined by their $\delta^{13}C$ and $\delta^{15}N$ isotopic values. Instead of using the individual species as groups, we formed the groups using a multivariable cluster analysis of their isotopic values with JMP 14. For further calculations it is necessary that the groups show a clear separation of the core niches (SEA), which we tested with the R package SIBER. To reconstruct the proportions of different prey group in the protein fraction of the carnivore diet, we used the R package MixSIAR (Bayesian Mixing Models in R, Stock and Semmens [66]). Initially, such Bayesian mixing models (e.g., MixSIAR, FRUITS, SIAR) were designed for ecologists who work with recent ecosystems and food chains but the model has been subsequently successfully applied to archaeological contexts [54, 67, 78–81].

MixSIAR allows the reconstruction of the most likely diet of the carnivores based on the nitrogen and carbon isotopes from their bone collagen relative to the isotopic values from their prey species. Essential for this calculation is the trophic enrichment factor (TEF) that quantifies the increase of $\delta^{13}C$ and $\delta^{15}N$ values in collagen from prey to predator. Indeed, the stable isotope composition of a predator differs from the composition of its prey in a predictable manner. The TEFs correspond to the difference between the stable isotope ratios of the consumer (predator collagen) and its diet (prey collagen) and are the result of the

discrimination of stable isotopes due to the behavior and physiology of the consumer [54, 62, 82]. For our study, we used the same TEF values ($\Delta^{13}C$ = 1.1 ± 1.1‰; $\Delta^{15}N$ = 3.2 ± 1.8‰)[62] for all carnivores. To get a robust statistical analysis, we set the MCMC (Markov Chain Monte Carlo, see Stock and Semmens [66]) chain length to 1,000,000 with a burn-in of 500,000 in 3 chains. Verification of the model convergence was done with Gelman-Rubin and Geweke tests (for detailed explanation, see Stock and Semmens [66]). In brief, the Gelman-Rubin test shows model convergence if the values are near 1. In most analyses, values below 1.1 are acceptable [83]. Additionally, the Geweke test compares the mean of the first part of each chain with the mean of the second part, using a two-sided z-test. If both means are the same, the model is convergent [66, 84].

All niche modeling (SIBER) and diet reconstructions (MixSIAR) were done using R Version 3.6.1.

## Results

### Elemental and isotopic analyses

The %$N_{bone}$ values measured on 71 carnivore samples and two hare samples confirmed the favorable conditions of preservation (0.6–3.8%), establishing quantitatively that collagen is preserved in the samples. Moreover, the atomic C:$N_{coll}$ ratios of all analyzed carnivores (3.2–3.5) and the small mammal samples (3.2–3.6), showed that the preservation of collagen was appropriate for the interpretation of the isotopic analysis in palaeobiological terms (Tables 1 and 2). Among the isotopic values, we found only minor difference between the average of wolves ($\delta^{13}C$ = -19.6 ± 0.6‰ and $\delta^{15}N$ = +9.1 ± 0.9‰), lynx ($\delta^{13}C$ = -19.3 ± 0.4‰ and $\delta^{15}N$ = +8.5 ± 1.2‰) and wolverines ($\delta^{13}C$ = -19.1 ± 0.0‰ and $\delta^{15}N$ = +8.5 ± 1.2‰). Compared to each other, red foxes ($\delta^{13}C$ = -20.0 ± 0.4‰ and $\delta^{15}N$ = +7.3 ± 1.9‰) and Arctic fox ($\delta^{13}C$ = -20.0 ± 0.8‰ and $\delta^{15}N$ = +7.6 ± 2.6‰) were also very similar in their average isotope values. In contrast to the other carnivores, both fox species yielded a slightly lower average $\delta^{13}C$ and $\delta^{15}N$ values and there was one outlier for each species among the Middle Palaeolithic specimens (PLC-76 and VLP-10).

The rodents average isotopic values of Arctic lemming ($\delta^{13}C$ = -21.3 ± 1.4‰ and $\delta^{15}N$ = +5.0 ± 2.6‰), Norway lemming ($\delta^{13}C$ = -22.5 ± 1.2‰ and $\delta^{15}N$ = +4.2 ± 2.4‰) and vole ($\delta^{13}C$ = -22.4 ± 0.5‰ and $\delta^{15}N$ = +5.6 ± 1.4‰) covered a wide range of the pre-LGM isospace, and were quite similar to each other. In contrast, the analyzed hare samples ($\delta^{13}C$ = -20.3 ± 0.2‰ and $\delta^{15}N$ = +3.2 ± 0.4‰) showed slightly higher $\delta^{13}C$ and lower $\delta^{15}N$ values than the lemming species and the voles.

### Trophic niche modeling

To form the pre-LGM isospace, we need to define the herbivore groups that are the assumed prey of most carnivores. Because with species-related groups the overlap of TA and SEA was too high, we built isotope-related groups (Fig 2). Finally, we generated four different prey groups, named after the most abundant taxon in the group (S3 Table). The "reindeer" group has the highest $\delta^{13}C$ values (-19.3 ± 0.5‰) and the lowest $\delta^{15}N$ values (+3.8 ± 0.7‰) and includes beside reindeer (n = 15) also hare (n = 2) and one Arctic lemming. The "horse" group includes horse (n = 19), Arctic lemming (n = 8) and Norway lemming (n = 4) and exhibits $\delta^{13}C$ values of -20.8 ± 0.3‰ and $\delta^{15}N$ values of +6.4 ± 0.7‰. The "mammoth" group shows $\delta^{13}C$ values of -21.1 ± 0.3‰ and the highest $\delta^{15}N$ values (+8.74 ± 0.5‰) and contains beside mammoth (n = 12) also two horses. Finally, the "rodent" group includes Norway lemming (n = 15), vole (n = 15), horse (n = 3) and Arctic lemming (n = 2) with $\delta^{13}C$ and $\delta^{15}N$ values of -22.5 ± 1.0‰ and +4.4 ± 2.3‰, respectively.

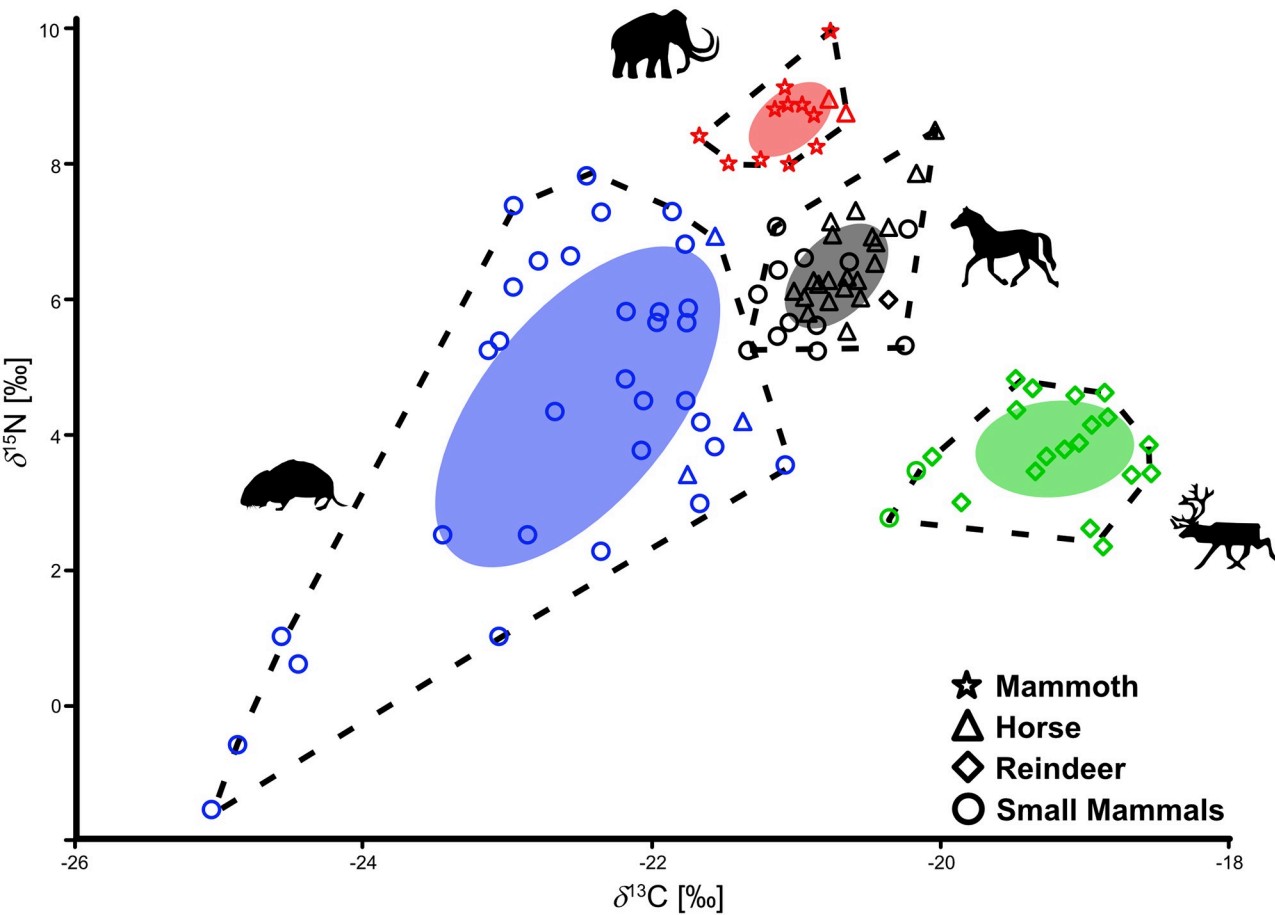

**Fig 2. Dietary sources in the isospace.** Dietary sources calculated with SIBER. Groups are named by the most abundant taxon. Dashed lines show the convex hull of the groups, while collard ellipses show the calculated Standard Ellipse Area (SEA).

For the carnivores we have primarily distinguished between foxes and large carnivores. Both fox species are combined here, as they differ very little in their isotope values (see chapter "Statistical test for isotopic variance of both fox species" in S2 Text). We were able to calculate three niche types of foxes: First foxes with high $\delta^{15}$N values (range from +7.1 to +10.0‰), then foxes with intermediate $\delta^{15}$N values (range from +3.7 to +6.7‰), and finally foxes with low $\delta^{15}$N values (range from +1.0 to +3.0‰). Additionally, the low $\delta^{15}$N foxes show lower $\delta^{13}$C values (range from -21.4 to -21.1‰) than the nearby intermediate $\delta^{15}$N fox group (range from -20.5 to -19.4‰). Finally, we also found these niche types in the respective periods, although not all niches at once. This results in a total of six fox niches and three large carnivore groups (Figs 3–5, S3 Table). For the Middle Palaeolithic, we found a large carnivore group ($\delta^{13}$C = -19.7 ± 0.4‰ and $\delta^{15}$N = +8.6 ± 1.2‰), a high $\delta^{15}$N fox niche ($\delta^{13}$C = -20.1 ± 0.3‰ and $\delta^{15}$N = +8.7 ± 0.8‰), and a low $\delta^{15}$N fox niche ($\delta^{13}$C = -21.3 ± 0.2‰ and $\delta^{15}$N = +2.0 ± 1.4‰). The Aurignacian is represented by a large carnivore group ($\delta^{13}$C = -19.5 ± 0.6‰ and $\delta^{15}$N = +9.3 ± 0.9‰), a high $\delta^{15}$N fox niche ($\delta^{13}$C = -19.9 ± 0.6‰ and $\delta^{15}$N = +8.6 ± 0.4‰), and an intermediate $\delta^{15}$N fox niche ($\delta^{13}$C = -20.0 ± 0.3‰ and $\delta^{15}$N = +5.4 ± 0.5‰). Finally, in the Gravettian, we calculated a large carnivore group ($\delta^{13}$C = -19.4 ± 0.7‰ and $\delta^{15}$N = +8.8 ± 0.8‰), a high $\delta^{15}$N fox niche ($\delta^{13}$C = -20.3 ± 0.3‰ and $\delta^{15}$N = +8.5 ± 1.1‰), and an intermediate $\delta^{15}$N fox niche ($\delta^{13}$C = -19.8 ± 0.5‰ and $\delta^{15}$N = +5.1 ± 1.5‰) as well. In order

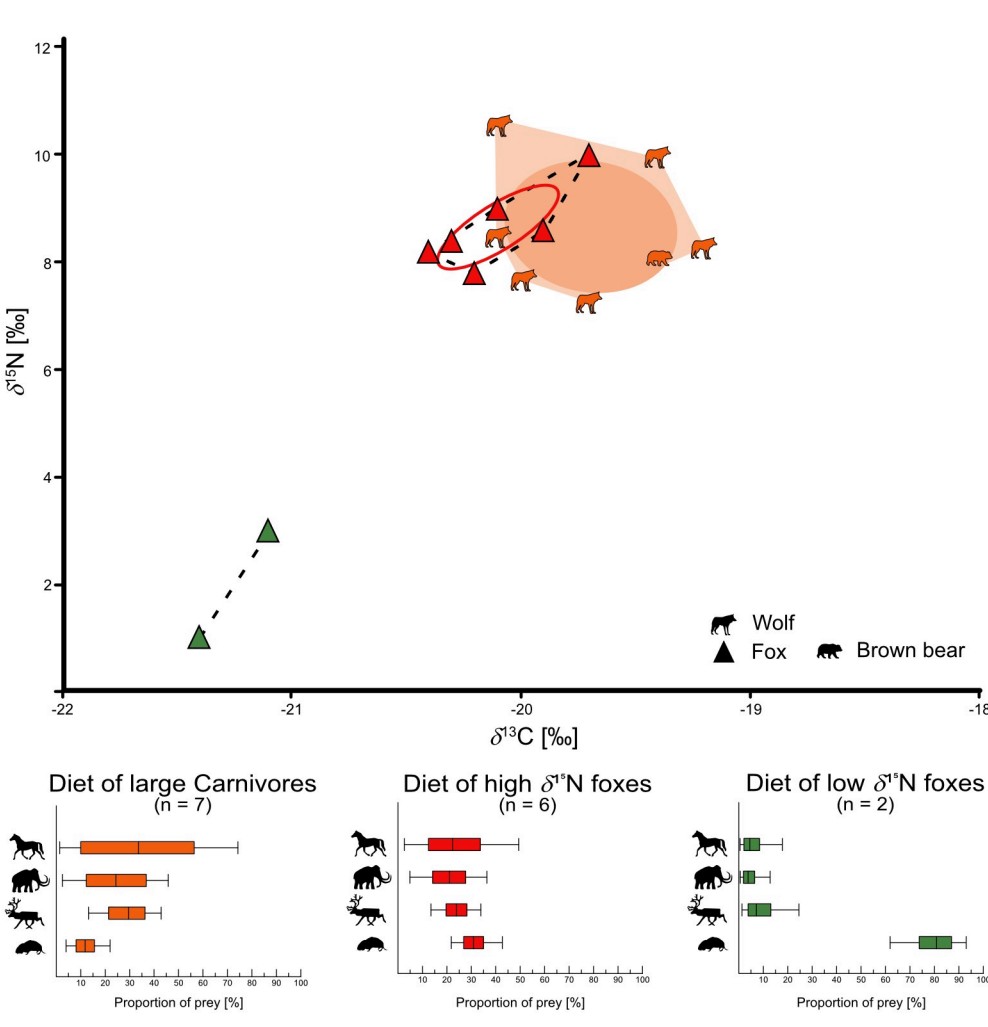

**Fig 3. Trophic niches in the Middle Palaeolithic.** Calculated trophic niches of foxes (Triangles) and large carnivores (shapes) from the Middle Palaeolithic with SIBER. Dashed lines in the fox niches as well as light collard area in the large carnivore group show the convex hull (outline of the niches). The solid lines in the fox niches and the dark collard ellipses in the carnivore group show the calculated Standard Ellipse Area (SEA) and reflect the core niches, based on Bayesian statistics. In the lower part of the figure is the reconstructed diet given. Diet proportions calculated with MixSIAR of each fox niche and the large carnivore group. Solid lines show the 5 to 95% confidence interval, full boxes show the 25 to 75% confidence interval and vertical black line shows the median value.

to indicate trophic niche competition or approaches of commensal behavior of foxes to other carnivores, we have calculated the SEAc overlap of the large carnivore group and the high $\delta^{15}$N foxes (Table 3, S3 Table for all calculated niche parameters). During all three periods the group of large carnivores overlaps with the high $\delta^{15}$N fox niches to a large extent (Middle Pal. = 49.6%, Aurignacian = 39.0%, Gravettian = 41.0%).

## Dietary reconstruction

The MixSIAR calculated model for dietary reconstructions showed convergence in both tests. Both diagnostics tested 127 variables of the model. In the Gelman-Rubin test, no variable was higher than 1.01. Additionally, the Geweke diagnostic revealed only three unequal variables in

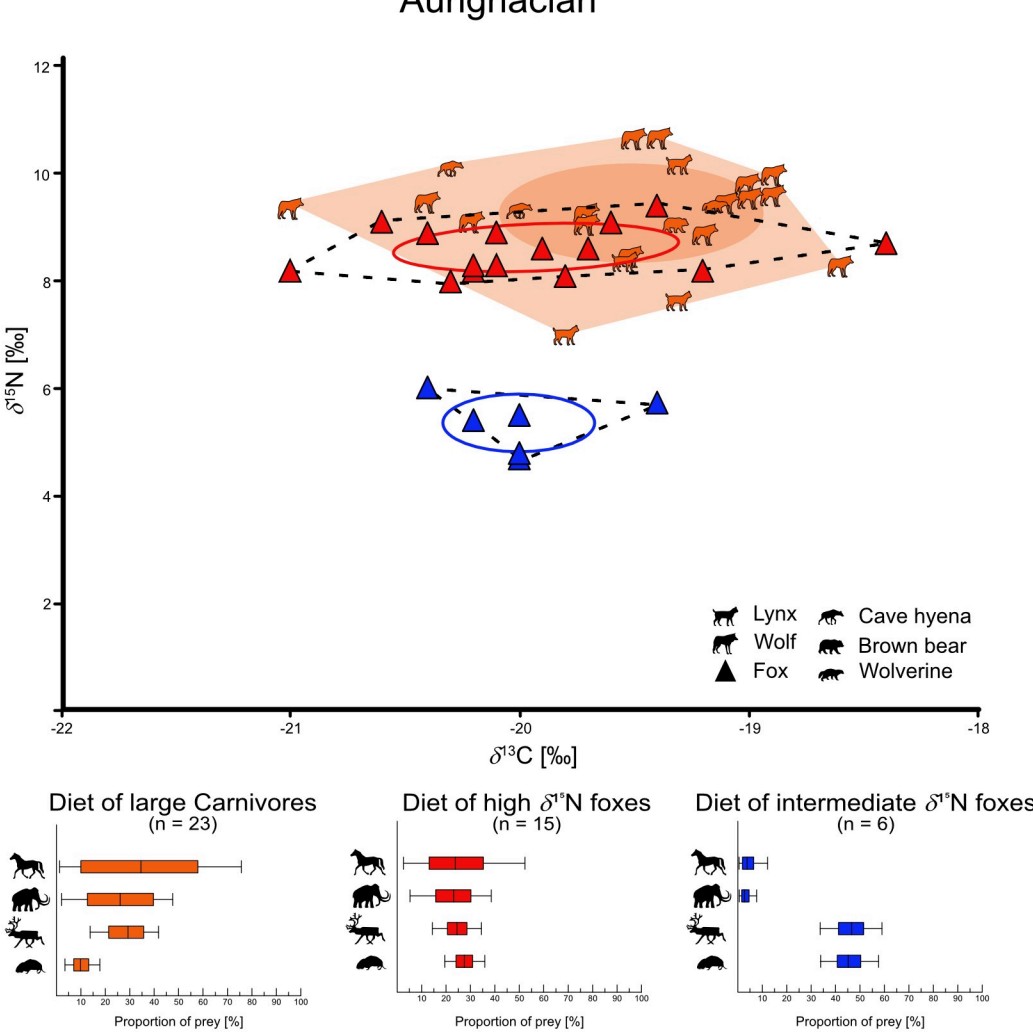

**Fig 4. Trophic niches in the Aurignacian.** Calculated trophic niches of foxes (Triangles) and large carnivores (shapes) from the Aurignacian with SIBER. Dashed lines in the fox niches as well as light collard area in the large carnivore group show the convex hull (outline of the niches). The solid lines in the fox niches and the dark collard ellipses in the carnivore group show the calculated Standard Ellipse Area (SEA) and reflect the core niches, based on Bayesian statistics. In the lower part of the figure is the reconstructed diet given. Diet proportions calculated with MixSIAR of each fox niche and the large carnivore group. Solid lines show the 5 to 95% confidence interval, full boxes show the 25 to 75% confidence interval and vertical black line shows the median value.

chain 1, nine unequal variables in chain 2 and three unequal variables in chain 3 out of 127. Therefore, the calculated model is usable for the dietary reconstruction.

We reconstructed the percentages of the four different prey sources for each fox niche as well as for the large carnivore groups as a whole and separated per period (Table 4, Figs 3–5). The dietary preferences of individuals varied strongly between the niches. Large carnivores of all periods preferred "horse" (34.6 ± 25%), "reindeer" (29.7 ± 9.7%) and "mammoth" (24.7 ± 14.3%). High $\delta^{15}$N foxes included all sources of analyzed prey in their diet in a similar proportion ("Rodents" = 29.2 ± 5.7%, "reindeer" = 25.1 ± 6.6%, "horse" = 24.1 ± 14.5% and "mammoth" = 21.5 ± 9.5%). However, intermediate $\delta^{15}$N foxes were more specialized on "rodents" (46.6 ± 8.5%) and "reindeer" (45.9 ± 8.9%), while low $\delta^{15}$N foxes fed primary on "rodents" (79.6 ± 9.8%).

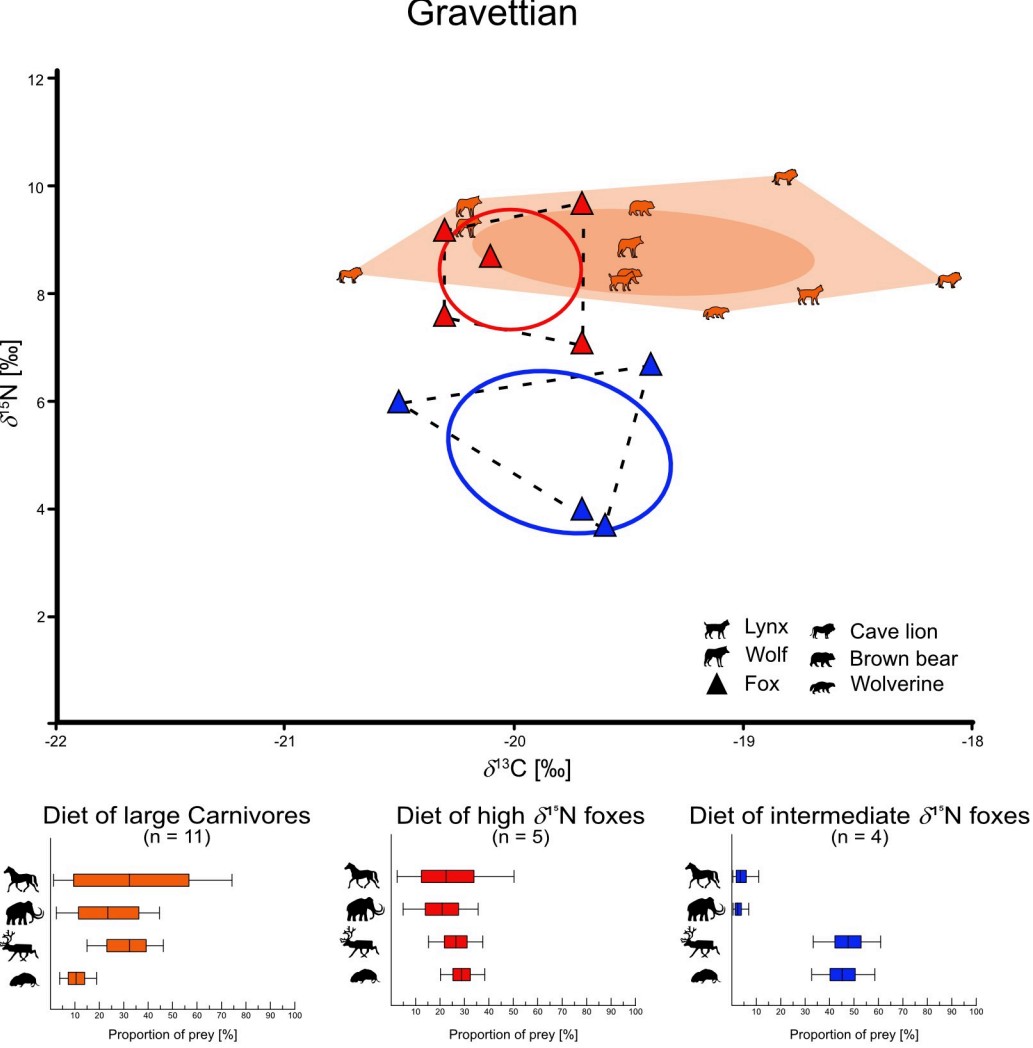

**Fig 5. Trophic niches in the Aurignacian.** Calculated trophic niches of foxes (Triangles) and large carnivores (shapes) from the Gravettian with SIBER. Dashed lines in the fox niches as well as light collard area in the large carnivore group show the convex hull (outline of the niches). The solid lines in the fox niches and the dark collard ellipses in the carnivore group show the calculated Standard Ellipse Area (SEA) and reflect the core niches, based on Bayesian statistics. In the lower part of the figure is the reconstructed diet given. Diet proportions calculated with MixSIAR of each fox niche and the large carnivore group. Solid lines show the 5 to 95% confidence interval, full boxes show the 25 to 75% confidence interval and vertical black line shows the median value.

## Discussion

For the three periods studied (Middle Palaeolithic, Aurignacian and Gravettian), we have sampled both large predators and foxes. Considering the large overlap of the high $\delta^{15}$N fox niches with the large carnivores in all periods together with the dietary reconstructions, we conclude that they consumed a similar diet, consisting of large mammals. However, the high $\delta^{15}$N foxes' diet was complemented by rodents. Intermediate $\delta^{15}$N foxes, primarily fed on reindeer and rodents, while low $\delta^{15}$N foxes fed almost exclusively on rodents.

The Late Pleistocene pre-LGM is a long period of time and covers several tens of thousands of years, so the question naturally arises as to whether the changes in the niches may also be due to environmental factors. Studies on the trophic niches of mammoths and horses have

**Table 3.** Calculated niche overlap between large carnivores and high $\delta^{15}$N foxes, based on SIBER.

| | Middle Palaeolithic | | |
|---|---|---|---|
| | Large Carnivores | High $\delta^{15}$N foxes | Overlap Area |
| TA [‰$^2$] | 2.08 | 0.46 | |
| SEA [‰$^2$] | 1.41 | 0.38 | |
| SEAc [‰$^2$] | 1.69 | 0.47 | 0.23 |
| %overlap | 13.8% | 49.6% | |
| | Aurignacian | | |
| | Large Carnivores | High $\delta^{15}$N foxes | Overlap Area |
| TA [‰$^2$] | 5.19 | 2.27 | |
| SEA [‰$^2$] | 1.57 | 0.83 | |
| SEAc [‰$^2$] | 1.65 | 0.89 | 0.35 |
| %overlap | 21.1% | 39.0% | |
| | Gravettian | | |
| | Large Carnivores | High $\delta$15N foxes | Overlap Area |
| TA [‰$^2$] | 4.20 | 1.26 | |
| SEA [‰$^2$] | 1.87 | 1.04 | |
| SEAc [‰$^2$] | 2.08 | 1.38 | 0.57 |
| %overlap | 27.2% | 41.0% | |

TA = Total Area (= convex hull), SEA = Standard Ellipse Area, SEAc = Standard Ellipse Area corrected to small sample size.

shown that environmental changes in the Swabian Jura played only a minor role within the early Upper Palaeolithic isospace[68, 79]. This is shown by the newly analyzed rodent isotope data as well, which are quite similar during the Aurignacian and Gravettian ($\delta^{13}$C: Aurignacian (n = 10) = -21.5 ± 0.7‰; Gravettian (n = 4) = -21.1 ± 0.6‰; t-test, $p$ = 0.35; $\delta^{15}$N: Aurignacian (n = 10) = +5.25 ± 1.2‰; Gravettian (n = 4) = +4.05 ± 1.4‰; t-test, $p$ = 0.19). More difficult to explain is the long period of the Middle Palaeolithic, resulting in a higher variability of rodent isotopes (n = 28; $\delta^{13}$C: -22.6 ± 1.1‰; $\delta^{15}$N: +4.8 ± 2.6‰). However, the high $\delta^{15}$N fox niche and the large carnivore group of the Middle Palaeolithic overlap to a high degree with the Aurignacian niches (niche overlap about 55 to 60%), suggesting similar conditions for the feeding possibilities of these predators across the Middle and Upper Palaeolithic. Therefore, in the following, we will ignore potential natural environmental conditions that could affect the isospace and focus on trophic behavioral reconstructions.

## Trophic behavior of foxes

Opportunistic small predators, such as red and Arctic foxes, are very adaptable in their diet and can therefore easily change their trophic behavior [8, 9, 11, 12, 14, 43–46, 48–50, 85, 86]. The same individuals that have hunted rodents alone for years can quickly adapt to scavenging and maintain this as a new trophic behavior as long as it is effective. Different trophic behaviors can also be seen in the dietary and niche reconstructions of the analyzed foxes from the Middle Palaeolithic and early Upper Palaeolithic layers of the Swabian Jura.

The majority of the sampled foxes fall into the high $\delta^{15}$N fox niche and show a strong overlap with the respective large carnivore groups. The high core niche (SEAc) overlap on the foxes' side (39.0–49.6%, Table 3), as well as the similarity in the calculated diet of both groups (Figs 3–5, Table 4) leads to the conclusion that the high $\delta^{15}$N foxes were commensal to large carnivores. This behavior can also be observed in modern red and Arctic foxes [8, 9, 11–13, 43,

**Table 4. Reconstructed dietary proportions for the different fox niches and large carnivore groups.**

| | | n | Dietary source | Mean ± SD | 2.5% | 5% | 25% | 50% | 75% | 95% | 97.5% |
|---|---|---|---|---|---|---|---|---|---|---|---|
| Large Carnivores | | 41 | Horse | **34.6 ± 25%** | 0.6% | 1.4% | 10.0% | **33.3%** | 56.9% | 74.3% | 78.5% |
| | | | Mammoth | **24.7 ± 14.3%** | 1.0% | 2.5% | 12.2% | **24.5%** | 37.5% | 46.3% | 48.2% |
| | | | Reindeer | **29.7 ± 9.7%** | 11.7% | 14.2% | 22.0% | **30.4%** | 37.2% | 44.1% | 46.7% |
| | | | Rodents | **11 ± 4.8%** | 2.8% | 3.7% | 7.6% | **10.7%** | 14.1% | 19.3% | 21.5% |
| | **Middle Pal.** | 7 | Horse | **34.5 ± 24.8%** | 0.6% | 1.4% | 10.0% | **33.6%** | 56.4% | 74.3% | 78.3% |
| | | | Mammoth | **24.4 ± 14.1%** | 1.1% | 2.5% | 12.3% | **24.4%** | 36.8% | 45.8% | 47.9% |
| | | | Reindeer | **28.9 ± 9.4%** | 11.0% | 13.2% | 21.4% | **29.6%** | 36.3% | 42.9% | 45.0% |
| | | | Rodents | **12.2 ± 5.5%** | 3.0% | 4.0% | 8.1% | **11.8%** | 15.6% | 22.0% | 24.1% |
| | **Aurignacian** | 23 | Horse | **35.2 ± 25.5%** | 0.5% | 1.3% | 10.0% | **34.6%** | 57.9% | 75.7% | 80.1% |
| | | | Mammoth | **26 ± 15%** | 0.9% | 2.1% | 12.7% | **26.1%** | 39.7% | 47.6% | 49.5% |
| | | | Reindeer | **28.6 ± 9%** | 11.1% | 13.8% | 21.4% | **29.3%** | 35.7% | 41.8% | 43.6% |
| | | | Rodents | **10.2 ± 4.3%** | 2.7% | 3.5% | 7.1% | **9.9%** | 13.2% | 17.8% | 19.1% |
| | **Gravettian** | 11 | Horse | **34.2 ± 25.1%** | 0.5% | 1.3% | 9.5% | **32.3%** | 56.7% | 74.3% | 77.6% |
| | | | Mammoth | **23.6 ± 13.8%** | 1.0% | 2.4% | 11.5% | **23.5%** | 36.2% | 44.6% | 46.4% |
| | | | Reindeer | **31.3 ± 10.1%** | 12.1% | 15.0% | 23.1% | **32.4%** | 39.2% | 46.2% | 47.9% |
| | | | Rodents | **10.9 ± 4.7%** | 2.8% | 3.8% | 7.3% | **10.5%** | 14.0% | 18.9% | 20.5% |
| | | **n** | **Prey source** | **Mean ± SD** | **2.5%** | **5%** | **25%** | **50%** | **75%** | **95%** | **97.5%** |
| High $\delta^{15}$N foxes | | 26 | Horse | **24.1 ± 14.5%** | 1.3% | 2.7% | 12.7% | **23.0%** | 34.2% | 50.2% | 55.4% |
| | | | Mammoth | **21.5 ± 9.5%** | 3.1% | 5.1% | 14.9% | **21.6%** | 28.4% | 36.7% | 38.7% |
| | | | Reindeer | **25.1 ± 6.6%** | 12.5% | 14.3% | 20.7% | **24.9%** | 29.5% | 35.7% | 37.8% |
| | | | Rodents | **29.2 ± 5.7%** | 18.6% | 20.6% | 25.4% | **29.0%** | 32.8% | 39.0% | 41.0% |
| | **Middle Pal.** | 6 | Horse | **23.8 ± 14.5%** | 1.2% | 2.6% | 12.5% | **22.3%** | 33.7% | 49.4% | 54.1% |
| | | | Mammoth | **21 ± 9.4%** | 2.6% | 4.9% | 14.2% | **21.1%** | 27.7% | 36.4% | 38.2% |
| | | | Reindeer | **23.9 ± 6.3%** | 11.5% | 13.5% | 19.7% | **23.9%** | 28.2% | 33.9% | 35.7% |
| | | | Rodents | **31.4 ± 6.4%** | 20.5% | 21.8% | 26.9% | **30.9%** | 35.0% | 42.7% | 46.0% |
| | **Aurignacian** | 15 | Horse | **25.1 ± 15.3%** | 1.3% | 2.6% | 13.1% | **23.8%** | 35.3% | 52.3% | 57.8% |
| | | | Mammoth | **22.8 ± 9.9%** | 3.0% | 5.3% | 15.8% | **23.1%** | 30.2% | 38.5% | 40.7% |
| | | | Reindeer | **24.5 ± 6%** | 12.7% | 14.4% | 20.6% | **24.5%** | 28.6% | 34.5% | 36.1% |
| | | | Rodents | **27.6 ± 5%** | 18.1% | 19.5% | 24.1% | **27.5%** | 30.9% | 35.9% | 37.6% |
| | **Gravettian** | 5 | Horse | **23.9 ± 14.8%** | 1.2% | 2.5% | 12.4% | **22.5%** | 33.9% | 50.3% | 55.3% |
| | | | Mammoth | **20.7 ± 9.3%** | 2.6% | 4.9% | 14.0% | **20.9%** | 27.6% | 35.6% | 37.8% |
| | | | Reindeer | **26.5 ± 6.8%** | 13.2% | 15.3% | 21.8% | **26.5%** | 31.1% | 37.5% | 39.7% |
| | | | Rodents | **28.9 ± 5.5%** | 18.4% | 20.2% | 25.3% | **28.8%** | 32.4% | 38.3% | 40.4% |
| | | **n** | **Prey source** | **Mean ± SD** | **2.5%** | **5%** | **25%** | **50%** | **75%** | **95%** | **97.5%** |
| Intermediate $\delta15$N foxes | | 10 | Horse | **4.4 ± 3.6%** | 0.2% | 0.5% | 1.8% | **3.5%** | 6.2% | 11.2% | 13.4% |
| | | | Mammoth | **3.1 ± 2.2%** | 0.4% | 0.5% | 1.5% | **2.6%** | 4.1% | 7.4% | 8.6% |
| | | | Reindeer | **45.9 ± 8.9%** | 27.9% | 31.3% | 40.1% | **46.1%** | 51.7% | 60.1% | 62.7% |
| | | | Rodents | **46.6 ± 8.5%** | 30.9% | 33.5% | 41.2% | **46.4%** | 52.0% | 60.5% | 64.4% |
| | **Aurignacian** | 6 | Horse | **4.7 ± 3.7%** | 0.2% | 0.5% | 1.9% | **3.8%** | 6.5% | 12.2% | 14.1% |
| | | | Mammoth | **3.3 ± 2.3%** | 0.4% | 0.6% | 1.6% | **2.8%** | 4.6% | 7.7% | 9.0% |
| | | | Reindeer | **46.5 ± 7.8%** | 31.2% | 33.7% | 41.2% | **46.6%** | 51.5% | 59.0% | 61.9% |
| | | | Rodents | **45.5 ± 7.3%** | 31.3% | 33.9% | 40.7% | **45.2%** | 50.3% | 57.6% | 60.1% |
| | **Gravettian** | 4 | Horse | **4.3 ± 3.4%** | 0.2% | 0.4% | 1.8% | **3.5%** | 5.9% | 11.0% | 13.1% |
| | | | Mammoth | **2.9 ± 2.1%** | 0.3% | 0.5% | 1.4% | **2.4%** | 3.9% | 6.9% | 8.1% |
| | | | Reindeer | **47.5 ± 8.4%** | 30.6% | 33.3% | 42.2% | **47.6%** | 52.9% | 60.9% | 63.4% |
| | | | Rodents | **45.3 ± 7.9%** | 30.0% | 32.7% | 40.2% | **45.2%** | 50.5% | 58.5% | 60.9% |
| | | **n** | **Prey source** | **Mean ± SD** | **2.5%** | **5%** | **25%** | **50%** | **75%** | **95%** | **97.5%** |

*(Continued)*

**Table 4.** (Continued)

| | | | | | | | | | | | |
|---|---|---|---|---|---|---|---|---|---|---|---|
| Low $\delta^{15}$N foxes | | 2 | Horse | **6.2 ± 5.8%** | 0.3% | 0.5% | 2.1% | **4.5%** | 8.5% | 17.8% | 21.3% |
| | | | Mammoth | **4.8 ± 4.1%** | 0.4% | 0.6% | 1.9% | **3.8%** | 6.5% | 12.8% | 15.4% |
| | | | Reindeer | **9.4 ± 7.5%** | 0.9% | 1.3% | 3.8% | **7.2%** | 13.1% | 24.6% | 28.5% |
| | | | Rodents | **79.6 ± 9.8%** | 57.7% | 61.9% | 73.8% | **80.8%** | 87.0% | 93.0% | 94.2% |
| | **Middle Pal.** | 2 | Horse | **6.2 ± 5.8%** | 0.3% | 0.5% | 2.1% | **4.5%** | 8.5% | 17.8% | 21.3% |
| | | | Mammoth | **4.8 ± 4.1%** | 0.4% | 0.6% | 1.9% | **3.8%** | 6.5% | 12.8% | 15.4% |
| | | | Reindeer | **9.4 ± 7.5%** | 0.9% | 1.3% | 3.8% | **7.2%** | 13.1% | 24.6% | 28.5% |
| | | | Rodents | **79.6 ± 9.8%** | 57.7% | 61.9% | 73.8% | **80.8%** | 87.0% | 93.0% | 94.2% |

44, 46–50, 86, 87] and is also suggested by morphological studies in Late Pleistocene red and Arctic foxes from Belgium [15, 16]. The dimensions of the lower carnassial indicated a higher carnivorous specialization in comparison with modern specimens, especially in Late Pleistocene Arctic foxes, whereas it was not so pronounced in Late Pleistocene red foxes [15]. However, the isotope values of our studied red and Arctic foxes did not show any significant difference (S1 Text), which is why we could not conclude that the two species had different diets.

With a predicted average diet of 79.6 ± 9.8% on rodents, the two foxes from the low $\delta^{15}$N fox niche are most likely to be what we would expect from foxes: rodent hunters. Indeed, rodents are also the main component of the diet of most modern foxes. This is especially true for red foxes [8, 9, 11–13, 44, 48–50, 86], but also for Arctic foxes [43, 47, 85]. Interestingly, only two of 38 sampled foxes were found with a rodent-dominated diet.

The intermediate $\delta^{15}$N foxes have an increased proportion of reindeer and rodents in their calculated diet (Figs 4 and 5, Table 4). A commensalism to (already sampled) large predators can be excluded, since none of the individuals has similar isotopic values, which should be the case with similar nutrition and thus commensalism [67]. However, as foxes are not able to hunt reindeer, they must have had the opportunity to feed regularly and over several years on reindeer carcasses to get the $\delta^{15}$N values that we observed in these foxes [53, 61, 62]. Since none of the other studied predators had developed a similar feeding strategy, it appears to be an exclusive trophic niche for these foxes.

In order to explain the trophic behavior of the intermediate $\delta^{15}$N foxes, we now look at the archaeological context of the sites. This group of foxes occurs exclusively in the early Upper Palaeolithic of the Swabian Jura, the Aurignacian and Gravettian periods. The zooarchaeological record indicates that reindeer and horse were among the most important game species for Middle and early Upper Palaeolithic hunter-gatherers [20, 24, 34, 38, 40–42, 70]. In addition, during the Aurignacian and Gravettian, a large number of mammoth remains were found, which were further processed [34, 42, 88]. Mammoths were not brought to the cave as a whole, but were butchered at the kill sites, while reindeer were brought to the site in their entirety and butchered there [42]. This behavior of Palaeolithic humans opened up two different feeding opportunities for foxes and other predators. On the one hand they had the possibility to scavenge from a high $\delta^{15}$N protein resource at the human (mammoth) kill sites, on the other hand they could scavenge from reindeer carcasses near the camp sites, i.e., the cave sites (for a more detailed explanation of the archaeological interpretation, see S3 Text: Archaeological interpretation). The dietary reconstructions of the Aurignacian fox niches show that these resources were effectively used, each from one niche. Moreover, cut marks on, for instance, two mandibles from Vogelherd Cave [70] show that foxes were exploited for meat and fur, both mandibles were sampled in this study as well and fall into the high $\delta^{15}$N fox niche (PLC-16) and the

intermediate $\delta^{15}$N fox niche (PLC-13), respectively. This demonstrates that there was a direct interaction of humans with foxes from both niches.

## Seasonality, targeted fox hunting and natural death in caves

There are different approaches to explain how and when foxes came into the sites. Three of them we would like to discuss in more detail in this section and compare them with our obtained results.

The first hypothesis is that Neanderthals and anatomically modern humans occupied the caves only during certain times or seasons [20, 34, 38, 70, 89]. Due to this discontinuous occupation behavior, many caves were alternately inhabited by humans and cave bears [35–37]. The foxes could therefore only have been hunted irregularly by humans. However, indicators of seasonal occupation can be strongly influenced by taphonomic processes [90]. This is especially true when the time depth is several thousand years. For the Middle Palaeolithic as well as for the early Upper Palaeolithic there are only very few clear indications of seasonal occupation and, more importantly, there is no evidence that completely excludes a longer occupation (see more details in chapter "Archaeological interpretation" in S3 Text). Better indicators of human occupation are the thickness of the archaeological horizons and the lithic artefact density, as described by Conard [23], for example. Both factors pointed to a stronger occupation during the early Upper Palaeolithic, and only a weak occupation during the Middle Palaeolithic.

When considering the results from the present study in this context, we have to keep in mind that seasonal or one-time events cannot be documented with isotope analysis from bone collagen [53, 61, 62]. The carbon and nitrogen isotopic values rather integrate a larger time period over several years and show the average nutrition of the last years of the specimen's life. Conversely, this also means that these specimens must have had access to the calculated diet for several years before they died. This is possible for members of high and low $\delta^{15}$N fox niches, which reflect natural trophic behavior. However, the intermediate $\delta^{15}$N foxes from the early Upper Palaeolithic do not show any known trophic behavior, as they had a very restricted diet based on reindeer and rodents. If humans were responsible for this restriction in the prey spectrum, it would mean that they must have done so for several years and not just seasonally.

Next hypothesis, we would like to discuss, is the targeted hunting of foxes. The number of fox remains in the Aurignacian increases abruptly and continues to rise in the Gravettian [17, 20]. For the first time, perforated fox teeth appear in the Aurignacian, and Münzel [38] described fox teeth as the second most important raw material for ornaments after ivory. The importance of the fox seems to have grown in the early Upper Palaeolithic, but does this also mean that foxes were targeted hunted? Baumann [17] discussed several possibilities of Palaeolithic fox hunting and concluded that they must have been hunted with baited traps. This hunting method is not likely to select foxes for certain trophic niches, furthermore, it is more likely that this method will also catch foxes with low $\delta^{15}$N values. However, if the traps were set up only near the occupied caves or at human kill sites, this could explain the selection for certain fox niches. Setting traps in the vicinity also has the advantage that they can be controlled more quickly and more regularly, which increases the success rate of fox hunting and reduces the risk of the trapped fox being consumed by another predator before the hunter collected its catch.

Finally, we would like to examine the hypothesis of a natural death of foxes in the caves. As already mentioned, the archaeological periods each contains several thousand years. During this time, the cave sites were not permanently occupied, although probably for longer than just a few seasons. Nevertheless, there is always the chance that foxes died in the caves without

human intervention. Especially for foxes from the two "natural" trophic niches in the Middle Palaeolithic we consider this scenario. In the early Upper Palaeolithic, the low $\delta^{15}$N fox niche is missing and therefore we have an intermediate $\delta^{15}$N fox niche each in Aurignacian and Gravettian, which cannot be naturally evolved. Although, we have direct evidence for fox exploitation in both niches, it can never be excluded that foxes also died naturally in the caves.

As we have shown, none of the presented hypotheses alone can sufficiently explain the observed trophic fox niches. Although each hypothesis can address certain aspects, it is not possible to include all observed results. Therefore, we now turn to the question whether foxes could indicate human population density in the past.

## Could foxes indicate human population density in the past?

While the two trophic behaviors of the low and high $\delta^{15}$N foxes represented natural feeding strategies not associated with humans, we hypothesize that the intermediate $\delta^{15}$N foxes had adapted to humans. As the main diet of foxes in both niches was rodents and reindeer, each almost 50%, a commensalism to large predators can be excluded, since none of the individuals show similar isotopic values.

However, there are three arguments that suggest a possible commensalism to humans.

1. Reindeer was also one of the main prey of humans during the early Upper Palaeolithic of the Swabian Jura [20, 24, 34, 38, 41, 42, 70, 91]. Niven [42] explained that the Aurignacian hunters of Vogelherd Cave carried reindeer in their entirety to the site and exploited them there. This process certainly caused some food waste dominated by reindeer remains that would not be present without the influence of humans and could have benefited the foxes living there.

2. The absence of large predators with similar isotopic values indicates that the resources that intermediate $\delta^{15}$N foxes consumed, was probably not accessible for large predators, although people hunted large predators, such as cave lions or wolves. It is likely that Palaeolithic humans tolerated foxes because they were harmless and thus the dietary resource was more continuously available to them over a significant period of time, resulting in their isotope values in bone collagen.

3. The exclusive occurrence of these foxes' trophic behavior in the early Upper Palaeolithic. Despite the high number of fox bones sampled, no intermediate $\delta^{15}$N foxes were found in the Middle Palaeolithic. We assume that the intermediate $\delta^{15}$N fox niche may be related to the population density of humans living in the region and their influence on the Pleistocene ecosystem (see more details in chapter "Archaeological interpretation" in S3 Text). The higher population density of humans probably also led to more frequent visits to the caves, and the food supply from food waste resulting from the butchering of reindeer was more constant over longer times. These circumstances provided for the first time a trophic niche for foxes that lived commensal to humans. Such synanthropic behavior has already been demonstrated in the Swiss Magdalenian site Kesslerloch [67] and is not unusual even in modern foxes [13, 14, 45, 92].

The hypothesis that certain trophic behavior of foxes can only be explained by the regular presence of humans could be applied to other Upper Palaeolithic sites as well (see Table 5 and S4 Text for more detailed information) and graphically implemented in Fig 6.

**Table 5. Commensal foxes in other archaeological sites.**

| Site or region | Time range | Associated period | With human associated prey near camp sites | Foxes commensal to | References |
|---|---|---|---|---|---|
| Swabian Jura (Germany) | 100 to 42 kyr cal BP | MP | Reindeer and horse | Large carnivores | This study |
| Swabian Jura (Germany) | 42 to 34 kyr cal BP | Aurignacian | Reindeer and horse | Large carnivores and humans | This study |
| Swabian Jura (Germany) | 34 to 30 kyr cal BP | Gravettian | Reindeer and horse | Large carnivores and humans | This study |
| Předmostí I (Czech Rep.) | 32 to 28.6 kyr cal BP | Gravettian | Reindeer | Large carnivores | Bocherens [54] |
| Buran-Kaya-III (Crimea) | 37 to 33 kyr cal BP | early UP | Saiga antelopes | Humans | Péan [93] |
| Swabian Jura (Germany) | 16.7 to 14 kyr cal BP | Magdalenian | Reindeer and horse | Large carnivores | Baumann [67] |
| Kesslerloch (Switzerland) | 16.7 to 14 kyr cal BP | Magdalenian | Reindeer and horse | Humans | Baumann [67] |

MP = Middle Palaeolithic, UP = Upper Palaeolithic. More detailed information in S4 Text.

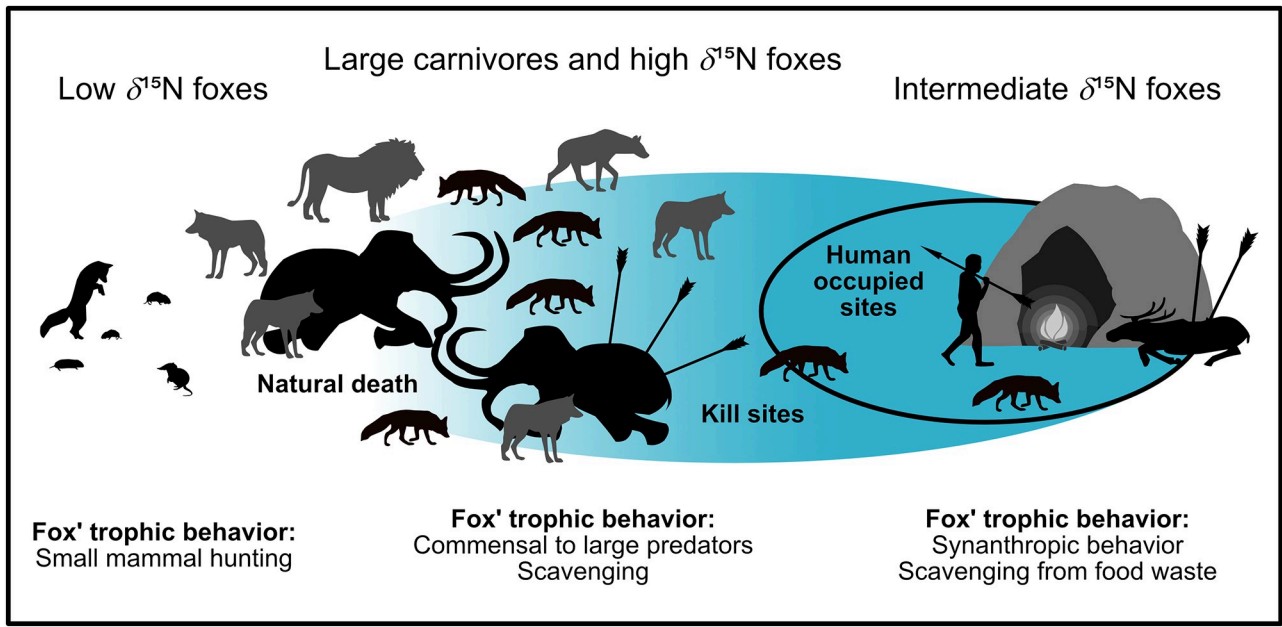

**Fig 6. Summary figure for the commensal fox hypothesis.** The blue area marks the impact of humans on dietary resources. For low $\delta^{15}$N foxes, humans had no influence, while for intermediate $\delta^{15}$N foxes they had a very strong influence (restricted diet). High $\delta^{15}$N foxes may be influenced (e.g. by scavenging at kill sites) or may be of natural origin (e.g. by scavenging from megafauna that died naturally).

## Conclusion

In this study we have shown how the trophic behavior of Pleistocene foxes changed from the Middle Palaeolithic to the early Upper Palaeolithic in the Swabian Jura. The majority of the sampled foxes exhibited high $\delta^{15}$N values, indicating commensal behavior to large carnivores, and were present in all periods studied. More interesting, however, has been the group of intermediate $\delta^{15}$N foxes, which had consumed a highly restricted diet on reindeer for several years before their death and only appeared in the early Upper Palaeolithic. These specimens may have fed on human food remains. The third group of foxes has low $\delta^{15}$N values and fed almost exclusively on rodents. However, this trophic niche could only be found in the Middle Palaeolithic of the Swabian Jura, which was sparsely populated by Neanderthals.

This leads us to our second goal in this study. We wanted to test to what extent foxes can be used as indicators of human population density and the resulting influence on the Pleistocene

ecosystem. With this study we were able to show that the influence of humans on the trophic behavior of small opportunistic predators, such as foxes, is quite recognizable in the Pleistocene. The commensal behavior of foxes to large carnivores, as well as the independent hunting of rodents, are natural trophic behaviors and also recognizable in modern foxes [8, 9, 11, 12, 14, 43–46, 48–50, 85, 86]. The two niches in the Aurignacian and Gravettian, respectively, which show a strongly restricted diet on reindeer and rodents, did not natural occurred and can be attributed to human influence, since reindeer was also a main prey of Paleolithic hunters and was often found in the zooarchaeological record [20, 24, 34, 38, 41, 42, 70, 91]. Even if our data only refer to a small region (Swabian Jura: Ach and Lone Valleys), we assume that our hypothesis can also be applied to other regions. Our results and conclusions agree with the human impacts on the Pleistocene ecosystem already determined by studies on mammoths (see also chapter "Possible impact of prehistoric people on Pleistocene ecosystems" in S5 Text).

For a better understanding of trophic niches and the interaction of foxes and humans during the Palaeolithic, besides further isotope analyses of such strongly human-influenced sites, it is also important to integrate sites that are not directly connected to humans or have only very low/irregular human occupation. This will help to gain a better understanding of the foxes' natural trophic niches and thus to better evaluate human influence. In the Swabian Jura, for example, these would be the two cave sites Fetzershaldenhöhle (Lone Valley; [94]) and Kogelstein (Ach Valley, [95]), which are distinguished as hyena dens, as well as Bärenhöhle (near Sonnenbühl-Erpfingen; [96]) and Schafstall (near Veringenstadt; [97–99]).

We expect that further methods, such as isotopic studies on individual amino acids, will lead to a more detailed dietary reconstruction and, based on this, to a differentiated consideration of trophic niches. This will then also lead to better interpretations of the human influence on Pleistocene foxes' niches. Further studies on strontium may also be useful, as this could provide information on the geographical position and movement patterns of foxes. Last but not least, with this study we have provided an impulse to pay a little more attention to small opportunistic predators as they may be the key to understanding human-made changes in Europe's Pleistocene ecosystems.

## Supporting information

**S1 Text. Statistical test for isotopic variance of both fox species**
(PDF)

**S2 Text. Intra-individual variability**
(PDF)

**S3 Text. Archaeological interpretation**
(PDF)

**S4 Text. Applying the hypothesis to other archaeological sites**
(PDF)

**S5 Text. Possible impact of prehistoric people on Pleistocene ecosystems**
(PDF)

**S1 Fig. Calculated trophic niches of foxes from the Middle Palaeolithic.** Dashed lines show the convex hull (outline of the niches), while the collard ellipses show the calculated Standard Ellipse Area (SEA) and reflect the core niches, based on Bayesian statistics. BS = Bockstein, HF = Hohle Fels, HS = Hohlenstein-Stadel.
(TIFF)

**S2 Fig. Calculated trophic niches of foxes from the Aurignacian.** Dashed lines show the convex hull (outline of the niches), while the collard ellipses show the calculated Standard Ellipse Area (SEA) and reflect the core niches, based on Bayesian statistics. Red area = high δ15N foxes, blue area = intermediate δ15N foxes, BS = Bockstein, GK = Geißenklösterle, HF = Hohle Fels, HS = Hohlenstein-Stadel, Si = Sirgenstein, VH = Vogelherd.
(TIFF)

**S3 Fig. Calculated trophic niches of foxes from the Gravettian.** Dashed lines show the convex hull (outline of the niches), while the collard ellipses show the calculated Standard Ellipse Area (SEA) and reflect the core niches, based on Bayesian statistics. Red area = high δ15N foxes, blue area = intermediate δ15N foxes, BS = Bockstein, GK = Geißenklösterle, Si = Sirgenstein.
(TIFF)

**S4 Fig. Intra-individual variation in the Aurignacian samples.** Black symbols show the potential affected samples from Vogelherd (VH), grey symbols show the potential affected samples from Hohlenstein-Stadel (HS) and white symbols show the potential affected samples from Sirgenstein (Si). Solid lines indicated most likely samples originated from one individual, according to the isotopic values and the limits. Dotted lines indicates a more unlikely origin from one specimen.
(TIFF)

**S5 Fig. Intra-individual variation in the Aurignacian samples.** Black symbols show the potential affected samples from Geißenklösterle (GK) and white symbols show the potential affected samples from Sirgenstein (Si). Solid lines indicated most likely samples originated from one individual, according to the isotopic values and the limits. Dotted lines indicates a more unlikely origin from one specimen.
(TIFF)

**S1 Table. Additional isotopic data.** Isotopic values from carnivores and herbivores from Swabian Jura sites, taken from literature.
(XLSX)

**S2 Table. Prey groups** List of the included prey taxa, sorted into prey groups.
(XLSX)

**S3 Table. Fox niches** List of the included carnivore taxa, sorted into fox niches and large carnivore groups.
(XLSX)

**S4 Table. Niche parameters.** Niche parameters of all niches (n>3) calculated with SIBER. TA = Total Area (= convex hull), SEA = Standard Ellipse Area, SEAc = Standard Ellipse Area corrected to small sample size.
(XLSX)

**S5 Table. List of probably affected samples.** AH = archaeological horizon, MNI = Minimum Number of Individuals. Bold names show the samples that are most likely to come from an individual, based on the isotope values that are below the limit.
(XLSX)

**S6 Table. Isotopic differences.** Differences in the isotopic values of the archaeological, zooarchaeological and genetic ambiguous samples. Values below the limits are marked in red,

values above 1 show a clear difference and are marked in blue. Values between both limits are marked yellow.
(XLSX)

## Acknowledgments

We want to thank our colleagues for their helpful support, including Susanne C. Münzel, Giulia Toniato, Christoph Wißing, Yumeko Tarusawa and Saskia Pfrengle (University of Tübingen). For supporting our collection and lab work, we are thankful to Sara Rhodes, Britt M. Starkovich, Angel Blanco-Lapaz, Peter Tung and the Zooarchaeology and Biogeology working groups (University of Tübingen), as well as, Christian Sommer for creating the map. Furthermore, we thank the team from the Laboratory of Chronology (Finnish Museum of Natural History), the team from the Institute of Environmental Science and Technology (Universitat Autònoma de Barcelona), Bernd Steinhilber from the Isotope Geochemistry Working Group (University of Tübingen) and Bernice Nisch from the Hydrogeochemisty Working Group (University of Tübingen) for their technical support in the elemental and isotopic analysis. For reviewing our manuscript, we want to thank the two anonymous reviewers.

## Author Contributions

**Conceptualization:** Chris Baumann, Hervé Bocherens.

**Data curation:** Chris Baumann, Dorothée G. Drucker.

**Formal analysis:** Chris Baumann.

**Funding acquisition:** Nicholas J. Conard.

**Investigation:** Chris Baumann.

**Methodology:** Chris Baumann.

**Project administration:** Hervé Bocherens.

**Resources:** Hervé Bocherens.

**Supervision:** Hervé Bocherens, Dorothée G. Drucker, Nicholas J. Conard.

**Visualization:** Chris Baumann.

**Writing – original draft:** Chris Baumann.

**Writing – review & editing:** Chris Baumann, Hervé Bocherens, Dorothée G. Drucker, Nicholas J. Conard.

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
