## [Decision Letter · Decision Letter 0]

10 Apr 2020

PONE-D-20-05722

Fox dietary ecology as a tracer of human impact

on Pleistocene ecosystems

PLOS ONE

Dear Mr. Baumann,

Thank you for submitting your manuscript to PLOS ONE. After careful consideration, we feel that it has merit but does not fully meet PLOS ONE’s publication criteria as it currently stands. Therefore, we invite you to submit a revised version of the manuscript that addresses the points raised during the review process.

Both reviewers felt that the overall approach and content of the paper was convincing, as do I.  However some significant issues were raised by the reviewers. Reviewer 1 wonders wonders about some key analytical issues (i.e., on specimens and inter- and intra-individual variability; and on age of individuals). Reviewer 2 raises some questions about the nature of the isotope study itself that need to be addressed, but also wonders about the potential of expanding on fox-human interactions. Both reivewers, particularly Reviewer 1, have specific questions about points raised in the paper. 

We would appreciate receiving your revised manuscript by May 25 2020 11:59PM. To enhance the reproducibility of your results, we recommend that if applicable you deposit your laboratory protocols in protocols.io, where a protocol can be assigned its own identifier (DOI) such that it can be cited independently in the future. For instructions see: http://journals.plos.org/plosone/s/submission-guidelines#loc-laboratory-protocols

We look forward to receiving your revised manuscript.

Kind regards,

Michael D. Petraglia, Ph.D.

Academic Editor

PLOS ONE

Journal Requirements:

2. In your manuscript, please provide additional information regarding the specimens used in your study. Ensure that you have reported specimen numbers and complete repository information, including museum name and geographic location.

For more information on PLOS ONE's requirements for paleontology and archaeology research, see https://journals.plos.org/plosone/s/submission-guidelines#loc-paleontology-and-archaeology-research.

3. We note that Figure 1 in your submission contains a map image which may be copyrighted. All PLOS content is published under the Creative Commons Attribution License (CC BY 4.0), which means that the manuscript, images, and Supporting Information files will be freely available online, and any third party is permitted to access, download, copy, distribute, and use these materials in any way, even commercially, with proper attribution. For these reasons, we cannot publish previously copyrighted maps or satellite images created using proprietary data, such as Google software (Google Maps, Street View, and Earth). For more information, see our copyright guidelines: http://journals.plos.org/plosone/s/licenses-and-copyright.

b).    You may seek permission from the original copyright holder of Figure 1 to publish the content specifically under the CC BY 4.0 license.

b).    If you are unable to obtain permission from the original copyright holder to publish these figures under the CC BY 4.0 license or if the copyright holder’s requirements are incompatible with the CC BY 4.0 license, please either i) remove the figure or ii) supply a replacement figure that complies with the CC BY 4.0 license. Please check copyright information on all replacement figures and update the figure caption with source information. If applicable, please specify in the figure caption text when a figure is similar but not identical to the original image and is therefore for illustrative purposes only.

Reviewers' comments:

Reviewer's Responses to Questions

**Comments to the Author**

1. Is the manuscript technically sound, and do the data support the conclusions?

Reviewer #1: Yes

Reviewer #2: Partly

2. Has the statistical analysis been performed appropriately and rigorously? 

Reviewer #1: I Don't Know

Reviewer #2: Yes

3. Have the authors made all data underlying the findings in their manuscript fully available?

Reviewer #1: Yes

Reviewer #2: Yes

4. Is the manuscript presented in an intelligible fashion and written in standard English?

Reviewer #1: Yes

Reviewer #2: Yes

5. Review Comments to the Author

Reviewer #1: Comments

The paper by Baumann et al. examines fox ecological niches throughout the Middle and Upper Palaeolithic of Europe and joins a growing corpus of research using stable isotopic analysis to elucidate the past diets of carnivores. The study provides an impressive new isotopic dataset from sites in the Swabian Jura, Germany, for a number of carnivore and herbivore taxa. Through time changes in fox ecological niches are examined in the context of changes in human populations. In summary, the authors conclude that the arrival of humans during the Upper Palaeolithic had a significant impact on regional ecology and opened up new ecological niches that were eventually filled by foxes. I believe the conclusions drawn by the authors to be sound and in line with ecological theory.

I have just a few major comments and a number of minor ones. My major concerns are related to the samples that make up the dataset. As it stands, it appears that some of the data may reflect intra-individual variability and it is unclear whether specimens may be isotopically enriched due to differences in age. These issues should be addressed before publication in PLoS One.

Major comments

It is unclear whether each specimen sampled can be confidently treated as a unique individual such that the resulting data represents strict inter-individual variability or it includes some intra-individual variability also. One example might be the two Canis lupus specimens PLC-37 and PLC-38 from the MPU archaeological horizon at the site of Hohlenstein-Stadel.

Were all the sampled specimens mature (adult) individuals? If young individuals are included in the dataset it is important to remember that they will be isotopically enriched from the consumption of their mother’s milk and that this enrichment will last until the juvenile bone has been remodeled.

The discussion section is long-winded and could be significantly reduced in some sections. For example, the paragraph starting line 553 could easily be reduced to one or two sentences.

Minor comments

Make sure the in-text referencing follows the PLoS One guidelines. For example, I assume that “Bocherens, Drucker (54)” doesn’t follow the journal guidelines.

Can you speculate as to why there are no low 15N foxes in the Upper Palaeolithic sites? Why might foxes have abandoned this ecological niche?

Line 22 – Can you make clear where the Swabian Jura is. This might not be obvious to some readers, including myself.

Line 42 – Evidence for hominin hunting at ~3.5 Ma, as far as I am aware, is non-existent. The earliest evidences for hominin consumption of meat dates to perhaps ~2.5 Ma. Early meat consumption was likely achieved via scavenging or, if by hunting, was infrequent. The earliest real evidence we see of persistent hominin carnivory dates to ~2.0 Ma (for example, see Ferraro et al., 2013. PLoS One 8: e62174).

Line 53 – Do you mean “prehistoric”?

Line 56 – Again it would be great to state where Swabian Jura is in the introduction.

Line 106 – ranges

Line 118 – Only five (5) carnivore species are listed here: wolf, brown bear, red fox, and arctic fox.

Line 231 – Were the same enrichment values applied to the other non-fox carnivores?

Line 253 – As per the comment above, is it possible that these outliers are younger/older than other individuals in your sample?

Line 257 – The difference in 13C values between hare and arctic lemming is actually less than the difference between the two lemming species. Likewise, the difference between 15N of hare on the one hand, and lemming and voles on the other, isn’t significantly different – i.e., there is only a 0.2% difference between the greatest difference of hare vs. non-hare and non-hare vs. non-hare.

Line 262 – I am curious as to how these groups were constructed? Just taking a quick glance at Fig. 2 it seems that many of the small mammals in the “horse group” could have easily been placed into the “small mammal group” (note the overlapping contour intervals).

Line 264 – These “species” groups are either not named after species (e.g., “rodent” group) or the named species is not the most common (e.g., “horse” group). For the latter, the most common taxon in the dataset is the genus Equus which includes horses, asses, and zebras. I recommend changing this to taxonomic groups, isotopic groups, or something else. Furthermore, as a reader I find these terms confusing for the dietary reconstruction (section 3.3). Particularly for the “horse” group which has almost an equal number of rodents. I wonder if there is a clearer and more informative system that could be employed?

Line 273 – It might be helpful to list the 15N ranges for each of the three niche categories. Also, under what rationale are the foxes grouped into the three nitrogen isotope groups? For example, why is PLC-73 with a 15N value of 3.7 included in the intermediate group and not the low group?

Line 275 – This line suggests that all niches (low, intermediate, and high 15N) are present in all three technological periods. I suggest rewording this.

Line 353 – I recommend changing the line “we conclude that they fed in a similar way”. There are huge differences in the feeding behavior of, say, foxes and hyenas.

Line 373 – Can you expand on the morphological study on the fox carnassial?

Line 379 – predators

Line 381 – This is a predicted diet not an average of the actual diet.

Line 441 – I recommend rewording this sentence. Something along the lines of “From the zooarchaeological record we know that fox remains are significantly more abundant in the Auriganican than the preceding…….”

Line 454 – The above comment about isotopic enrichment in young individuals is pertinent to assessing the validity of statements such as this.

Line 463 – What does a “one sided” diet mean?

Line 481 – How is the Gravettian, a specified technocomplex, identified as such by radiocarbon dating?

Line 488 – Is this meant to say the “high 15N foxes”?

Line 495 – Is this reference to human kill sites? Because kill sites include predation by any animal (including humans).

Line 526 – second?

Line 532 – In the future it might be worth looking to strontium isotopes to test this hypothesis.

Line 564 – What are the three niches of category A? Does this refer to the three archaeological periods (MP, Aurignacian, & Gravettian)? If so, can these be considered different niches just because they occur different times? Horses are considered to have occupied a similar niche for hundreds of thousands of years throughout the Plio-Pleistocene. I think revisiting how these different dietary and temporal niches are presented in the paper would be worthwhile. Lastly, I find this paragraph particularly difficult to follow and recommend rewriting it.

Line 586 – I’m not sure I buy the idea that large carnivores not being in the vicinity. At the very least, it seems likely that the home ranges of foxes in the vicinity of humans overlapped with some other larger carnivores in the broader region and, therefore, could have had access to larger carnivore refuse.

Line 645 – Are you able to present this graphically, perhaps in comparison to your own Swabian Jura data?

Line 652 – “which also often show….”

Line 687 – What possible role did changes in climate play in restructuring ecology and altering large mammal tropic niches?

Reviewer #2: The authors present interesting new stable isotope data for a range of large and small bodies mammals. The isotopic data complement existing faunal datasets, and the dietary modeling results contribute to the understanding of late Pleistocene trophic interactions.

I may have missed it, but prior to combining the data from two fox species, do the authors confirm that the d13C and d15N values are statistically indistinguishable?

Along the same lines, did the authors test for the effect of location among fox values within each time period? As well, most of the small mammal data are from a single site (HF), are other there data that support the use of these in dietary reconstruction for carnivores from other sites? Based on the map, the locations are not far apart, but I was curious if there is specific rationale for combining datasets across locations?

Are the fox specimens (n=9) in the human-influenced dietary category B from horizons with evidence of human occupation? Wondering if there are other data to support a fox-human interaction for these individuals beyond just the estimated high proportion of reindeer in their diet? The fox-human interaction conclusion would be strengthened by information from multiple lines of evidence.

The paper is quite long, I found it a bit challenging to locate and extract information, and follow the authors’ arguments and chains of logic. Perhaps the manuscript could benefit from a more concise format.

6. PLOS authors have the option to publish the peer review history of their article (what does this mean?). If published, this will include your full peer review and any attached files.

Reviewer #1: No

Reviewer #2: No

---

## [Author Response · Author response to Decision Letter 0]

27 May 2020

All responses are included in the file "Response to Reviewers".

---

## [Decision Letter · Decision Letter 1]

22 Jun 2020

Fox dietary ecology as a tracer of human impact

on Pleistocene ecosystems

PONE-D-20-05722R1

Dear Dr. Baumann,

We’re pleased to inform you that your manuscript has been judged scientifically suitable for publication and will be formally accepted for publication once it meets all outstanding technical requirements.

Kind regards,

Michael D. Petraglia, Ph.D.

Academic Editor

PLOS ONE

Reviewers' comments:

Line 452-453 – I believe that this is meant to read "...the second most important raw material...". Also, important to remember that stone is a raw material, and I doubt fox teeth specifically are more important.

Line 469 – when saying "natural origin" are you referring to tropic niche or natural death in the cave. I read it to mean as a natural death, which, if I'm correct, is at odds with the rest of the paragraph. Maybe try rewording it.

Line 495 - "exclusive"

Line 543 - "occur"

**Comments to the Author**

1. If the authors have adequately addressed your comments raised in a previous round of review and you feel that this manuscript is now acceptable for publication, you may indicate that here to bypass the “Comments to the Author” section, enter your conflict of interest statement in the “Confidential to Editor” section, and submit your "Accept" recommendation.

Reviewer #1: All comments have been addressed

2. Is the manuscript technically sound, and do the data support the conclusions?

Reviewer #1: Yes

3. Has the statistical analysis been performed appropriately and rigorously? 

Reviewer #1: I Don't Know

4. Have the authors made all data underlying the findings in their manuscript fully available?

Reviewer #1: Yes

5. Is the manuscript presented in an intelligible fashion and written in standard English?

Reviewer #1: Yes

6. Review Comments to the Author

Reviewer #1: The authors have addressed, where possible, all of the my comments. I think this study is ready for publication in PLOS ONE. I have just a few very minor comments

Line 452-453 – I believe that this is meant to read "...the second most important raw material...". Also, important to remember that stone is a raw material, and I doubt fox teeth specifically are more important.

Line 469 – when saying "natural origin" are you referring to tropic niche or natural death in the cave. I read it to mean as a natural death, which, if I'm correct, is at odds with the rest of the paragraph. Maybe try rewording it.

Line 495 - "exclusive"

Line 543 - "occur"

7. PLOS authors have the option to publish the peer review history of their article (what does this mean?). If published, this will include your full peer review and any attached files.

Reviewer #1: No

---

## [Editor Report · Acceptance letter]

29 Jun 2020

PONE-D-20-05722R1 

Fox dietary ecology as a tracer of human impact
on Pleistocene ecosystems 

Dear Dr. Baumann:

I'm pleased to inform you that your manuscript has been deemed suitable for publication in PLOS ONE. Congratulations! Your manuscript is now with our production department. 

Kind regards, 

on behalf of

Professor Michael D. Petraglia 

Academic Editor

PLOS ONE